# Evaluating Robustness and Uncertainty of Graph Models Under Structural Distributional Shifts

**Gleb Bazhenov**[*]
HSE University, Yandex Research

**Denis Kuznedelev**
Yandex Research, Skoltech

**Andrey Malinin**[†]
Isomorphic Labs

**Artem Babenko**
Yandex Research, HSE University

**Liudmila Prokhorenkova**
Yandex Research

## Abstract

In reliable decision-making systems based on machine learning, models have to be robust to distributional shifts or provide the uncertainty of their predictions. In node-level problems of graph learning, distributional shifts can be especially complex since the samples are interdependent. To evaluate the performance of graph models, it is important to test them on diverse and meaningful distributional shifts. However, most graph benchmarks considering distributional shifts for node-level problems focus mainly on node features, while structural properties are also essential for graph problems. In this work, we propose a general approach for inducing diverse distributional shifts based on graph structure. We use this approach to create data splits according to several structural node properties: *popularity*, *locality*, and *density*. In our experiments, we thoroughly evaluate the proposed distributional shifts and show that they can be quite challenging for existing graph models. We also reveal that simple models often outperform more sophisticated methods on the considered structural shifts. Finally, our experiments provide evidence that there is a trade-off between the quality of learned representations for the base classification task under structural distributional shift and the ability to separate the nodes from different distributions using these representations.

## 1 Introduction

Recently, much effort has been put into creating decision-making systems based on machine learning for various high-risk applications, such as financial operations, medical diagnostics, autonomous driving, *etc.* These systems should comprise several important properties that allow users to rely on their predictions. One such property is *robustness*, the ability of an underlying model to cope with *distributional shifts* when the features of test inputs become different from those encountered in the training phase. At the same time, if a model is unable to maintain high performance on a shifted input, it should signal the potential problems by providing some measure of *uncertainty*. These properties are especially difficult to satisfy in the node-level prediction tasks on graph data since the elements are interdependent, and thus the distributional shifts may be even more complex than for the classic setup with i.i.d. samples.

To evaluate graph models in the node-level problems, it is important to test them under diverse, complex, and meaningful distributional shifts. Unfortunately, most existing graph datasets split the nodes into train and test parts uniformly at random. Rare exceptions include the Open Graph Benchmark (OGB) [12] that creates more challenging non-random data splits by dividing the nodes

---

[*]Correspondence to `gv-bazhenov@yandex-team.ru`
[†]Work done while at Yandex Research

according to some domain-specific property. For instance, in the OGB-Arxiv dataset, the papers are divided according to their publication date, which is a realistic setup. Unfortunately, not many datasets contain such meta-information that can be used to split the data. To overcome this issue, Gui et al. [10] propose the Graph OOD Benchmark (GOOD) designed specifically for evaluating graph models under distributional shifts. The authors distinguish between two types of shifts: *concept* and *covariate*. However, creating such shifts is non-trivial, while both types of shifts are typically present simultaneously in practical applications. Also, the splitting strategies in GOOD are mainly based on the node features and do not take into account the graph structure.[3]

In our work, we fill this gap and propose a universal method for inducing *structural* distributional shifts in graph data. Our approach allows for creating diverse, complex, and meaningful node-level shifts that can be applied to any graph dataset. In particular, we introduce the split strategies that focus on such node properties as *popularity*, *locality*, and *density*. Our framework is flexible and allows one to easily extend it with other structural shifts or vary the fraction of nodes available for training and testing. We empirically show that the proposed distributional shifts are quite challenging for existing graph methods. In particular, the locality-based shift appears to be the most difficult in terms of the predictive performance for most considered OOD robustness methods, while the density-based shift is extremely hard for OOD detection by uncertainty estimation methods. Our experiments also reveal that simple models often outperform more sophisticated approaches on structural distributional shifts. In addition, we investigate some modifications of graph model architectures that may improve their OOD robustness or help in OOD detection on the proposed structural shifts. Our experiments provide evidence that there is a trade-off between the quality of learned representations for base classification task under structural distributional shift and the ability to separate the nodes from different distributions using these representations.

## 2 Background

### 2.1 Graph problems with distributional shifts

Several research areas in graph machine learning investigate methods for solving node-level prediction tasks under distributional shifts, and they are primarily different in what problem they seek to overcome. One such area is *adversarial robustness*, which requires one to construct a method that can handle artificial distributional shifts that are induced as adversarial attacks for graph models in the form of perturbed and contaminated inputs. The related approaches often focus on designing various data augmentations via introducing random or learnable noise [30, 42, 40, 35, 22].

Another research area is *out-of-distribution generalization*. The main task is to design a method that can handle real-world distributional shifts on graphs and maintain high predictive performance across different OOD environments. Several invariant learning and risk minimization techniques have been proposed to improve the robustness of graph models to such real-world shifts [1, 19, 20, 38].

There is also an area of *uncertainty estimation*, which covers various problems. In *error detection*, a model needs to provide the uncertainty estimates that are consistent with prediction errors, *i.e.*, assign higher values to potential misclassifications. In the presence of distributional shifts, the uncertainty estimates can also be used for *out-of-distribution detection*, where a model is required to distinguish the shifted OOD data from the ID data [24, 36, 2, 27, 18].

The structural distributional shifts proposed in our paper can be used for evaluating OOD generalization, OOD detection, and error detection since they are designed to replicate the properties of realistic graph data.

### 2.2 Uncertainty estimation methods

Depending on the source of uncertainty, it is usually divided into *data uncertainty*, which describes the inherent noise in data due to the labeling mistakes or class overlap, and *knowledge uncertainty*, which accounts for the insufficient amount of information for accurate predictions when the distribution of the test data is different from the training one [5, 24, 23].

---

[3]In Appendix C, we provide a detailed discussion of the GOOD benchmark.

**General-purpose methods** The most simple approaches are standard classification models that predict the parameters of softmax distribution. For these methods, we can define the measure of uncertainty as the entropy of the predictive categorical distribution. This approach, however, does not allow us to distinguish between data and knowledge uncertainties, so it is usually inferior to other methods discussed below.

*Ensembling techniques* are powerful but expensive approaches that providing decent predictive performance and uncertainty estimation. The most common example is *Deep Ensemble* [21], which can be formulated as an empirical distribution of model parameters obtained after training several instances of the model with different random seeds for initialization. Another way to construct an ensemble is *Monte Carlo Dropout* [6], which is usually considered as the baseline in uncertainty estimation literature. However, it has been shown that this technique is commonly inferior to deep ensembles, as the obtained predictions are dependent and thus lack diversity. Importantly, ensembles allow for a natural decomposition of total uncertainty in data and knowledge uncertainty [23].

There is also a family of *Dirichlet-based methods*. Their core idea is to model the point-wise Dirichlet distribution by predicting its parameters for each input individually using some model. To get the parameters of the categorical distribution, one can normalize the parameters of the Dirichlet distribution by their sum. This normalization constant is called *evidence* and can be used to express the general confidence of a Dirichlet-based model. Similarly to ensembles, these methods are able to distinguish between data and knowledge uncertainty. There are numerous examples of Dirichlet-based methods, one of the first being *Prior Network* [24] that induces the behavior of Dirichlet distribution by contrastive learning against the OOD samples. Although this method is theoretically sound, it requires knowing the OOD samples in the training stage, which is a significant limitation. Another approach is *Posterior Network* [2], where the behavior of the Dirichlet distribution is controlled by *Normalizing Flows* [15, 13], which estimate the density in latent space and reduce the Dirichlet *evidence* in the regions of low density without using any OOD samples.

**Graph-specific methods** Recently, several uncertainty estimation methods have been designed specifically for node-level problems. For example, *Graph Posterior Network* [33] is an extension of the *Posterior Network* framework discussed above. To model the Dirichlet distribution, it first encodes the node features into latent representations with a graph-agnostic model and then uses one flow per class for density estimation. Then, the *Personalized Propagation* scheme [17] is applied to the Dirichlet parameters to incorporate the network effects. Another Dirichlet-based method is *Graph-Kernel Dirichlet Estimation* [41]. In contrast to *Posterior Networks*, its main property is a compound training objective, which is focused on optimizing the node classification performance and inducing the behavior of the Dirichlet distribution. The former is achieved via knowledge distillation from a teacher GNN, while the latter is performed as a regularisation against the prior Dirichlet distribution computed via graph kernel estimation. However, as shown by Stadler et al. [33], this approach is inferior to *Graph Posterior Network* while having a more complex training procedure and larger computational complexity.

## 2.3 Methods for improving robustness

Improving OOD robustness can be approached from different perspectives, which, however, share the same idea of learning the representations that are invariant to undesirable changes of the input distribution. For instance, the main claim in *domain adaptation* is that to achieve an effective domain transfer and improve the robustness to distributional shift, the predictions should depend on the features that can not discriminate between the source and target domains. Regarding the branch of *invariant learning*, these methods decompose the input space into different environments and focus on constructing robust representations that are insensitive to their change.

**General-purpose methods** A classic approach for domain adaptation is *Domain-Adversarial Neural Network* [7] that promotes the emergence of features that are discriminative for the main learning task on the source domain but do not allow to detect the distributional shift on other domains. This is achieved by jointly optimizing the underlying representations as well as two predictors operating on them: the main label predictor that solves the base classification task and the domain classifier that discriminates between the source and the target domains during training. Another simple technique in the class of unsupervised domain adaptation methods is *Deep Correlation Alignment*

[34], which trains to align the second-order statistics of the source and target distributions that are produced by the activation layers of a deep neural model that solves the base classification task.

A representative approach in invariant learning is *Invariant Risk Minimization* [1], which searches for data representations providing decent performance across all environments, while the optimal classifier on top of these representations matches for all environments. Another method is *Risk Extrapolation* [20], which targets both robustness to covariate shifts and invariant predictions. In particular, it targets the forms of distributional shifts having the largest impact on performance in training domains.

**Graph-specific methods**   More recently, several graph-specific methods for improving OOD robustness have been proposed. One of them is an invariant learning technique *Explore-to-Extrapolate Risk Minimization* [38], which leverages multiple context explorers that are specified by graph structure editors and adversarially trained to maximize the variance of risks across different created environments. There is also a simple data augmentation technique *Mixup* [39] that trains neural models on convex combinations of pairs of samples and their corresponding labels. However, devising such a method for solving the node-level problems is not straightforward, as the inputs are connected to each other. Based on this technique, Wang et al. [37] have designed its adaptation for graph data: instead of training on combinations of the initial node features, this method exploits the intermediate representations of nodes and their neighbors that are produced by graph convolutions.

# 3   Structural distributional shifts

## 3.1   General approach

As discussed above, existing datasets for evaluating the robustness and uncertainty of node-level problems mainly focus on feature-based distributional shifts. Here, we propose a universal approach that produces non-trivial yet reasonable structural distributional shifts. For this purpose, we introduce a node-level graph characteristic $\sigma_i$ and compute it for every node $i \in \mathcal{V}$. We sort all nodes in ascending order of $\sigma_i$ — those with the smallest values of $\sigma_i$ are considered to be ID, while the remaining ones are OOD. As a result, we obtain a graph-based distributional shift where ID and OOD nodes have different structural properties. The type of shift depends on the choice of $\sigma_i$, and several possible options are described in Section 3.2 below.

We further split the ID nodes uniformly at random into the following parts:

- `Train` contains nodes $\mathcal{V}_{\text{train}}$ that are used for regular training of models and represent the only observations that take part in gradient computation.

- `Valid-In` enables us to monitor the best model during the training stage by computing the validation loss for nodes $\mathcal{V}_{\text{valid-in}}$ and choose the best checkpoint.

- `Test-In` is used for testing on the remaining ID nodes $\mathcal{V}_{\text{test-in}}$ and represents the simplest setup that requires a model to reproduce in-distribution dependencies.

The remaining OOD nodes are split into `Valid-Out` and `Test-Out` subsets based on their $\sigma_i$:

- `Test-Out` is used to evaluate the robustness of models to distributional shifts. It consists of nodes with the largest values of $\sigma_i$ and thus represents the most shifted part $\mathcal{V}_{\text{test-out}}$.

- `Valid-Out` contains OOD nodes with smaller values of $\sigma_i$ and thus is less shifted than `Test-Out`. This subset $\mathcal{V}_{\text{valid-out}}$ can be used for monitoring the model performance on a shifted distribution. Our experiments assume a more challenging setup when such shifted data is unavailable during training. However, the presence of this subset allows us to further separate $\mathcal{V}_{\text{test-out}}$ from $\mathcal{V}_{\text{train}}$ — the larger $\mathcal{V}_{\text{valid-out}}$ we consider, the more significant distributional shift is created between the train and OOD test nodes.

Our general framework is quite flexible and allows one to easily vary the size of the training part and the type of distributional shift. Let us now discuss some particular shifts that we propose in this paper.

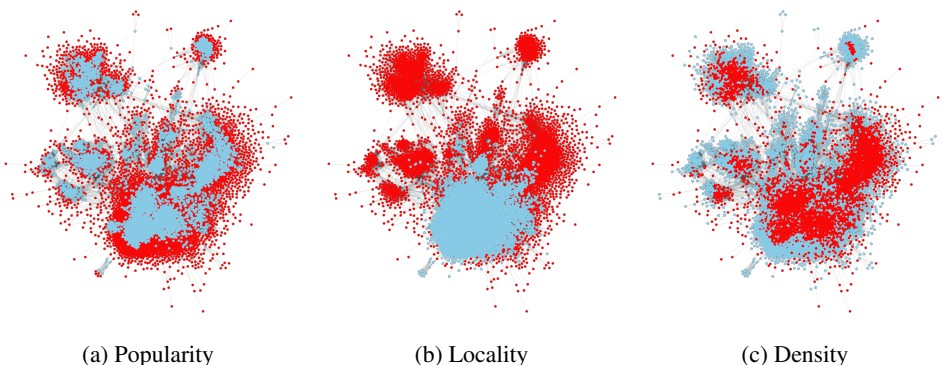

|(a) Popularity|(b) Locality|(c) Density|

Figure 1: Visualization of structural shifts for **AmazonPhoto** dataset: ID is blue, OOD is red.

## 3.2 Proposed distributional shifts

To define our data splits, we need to choose a node property $\sigma_i$ as a splitting factor. We consider diverse graph characteristics covering various distributional shifts that may occur in practice. In a standard non-graph ML setup, shifts typically happen only in the feature space (or, more generally, the joint distribution of features and targets may shift). However, in graph learning tasks, there can be shifts specifically related to the graph structure. We discuss some representative examples below.

**Popularity-based**  The first strategy represents a possible bias towards popularity. In some applications, it is natural to expect the training set to consist of more popular items. For instance, in the web search, the importance of pages in the internet graph can be measured via PageRank [29]. For this application, the labeling of pages should start with important ones since they are visited more often. Similar situations may happen for social networks, where it is natural to start labeling with the most influential users, or citation networks, where the most cited papers should be labeled first. However, when applying a graph model, it is essential to make accurate predictions on less popular items. Motivated by that, we introduce a popularity-based split based on PageRank. The vector of PageRank values $\pi_i$ describes the stationary distribution of a random walk with restarts, for which the transition matrix is defined as the normalized adjacency matrix $\mathbf{AD}^{-1}$ and the probability of restart is $\alpha$:

$$\boldsymbol{\pi} = (1 - \alpha)\mathbf{AD}^{-1}\boldsymbol{\pi} + \alpha\boldsymbol{p}. \qquad (1)$$

The vector of restart probabilities $\boldsymbol{p}$ is called a personalization vector, and $p_i = 1/n$ by default, which describes the uniform distribution over nodes.

To construct a popularity-based split, we use PageRank (PR) as a measure of node importance. Thus, we compute PR for every node $i$ and set $\sigma_i = -\pi_i$, which means that the nodes with smaller PR values (*i.e.*, less important ones) belong to the OOD subsets. Note that instead of PR, one can potentially use any other measure of node importance, *e.g.*, node degree, or betweenness centrality.

Figure 1a illustrates that the proposed shift separates the most important nodes that belong to the cores of large clusters and the structural periphery, which consists of less important nodes in terms of their PR values. We also observe that such popularity-based split agrees well with some natural distributional shifts. For instance, Figure 11a in Appendix shows the distribution of PageRank in train and test parts of the **OGB-Arxiv** dataset [12]. Here, the papers are split according to their publication date. Thus, older papers naturally have more citations and, therefore, larger PageRank. Our proposed split allows one to mimic this behavior for datasets without timestamps available. We refer to Appendix A for more details and additional analysis.

**Locality-based**  Our next strategy is focused on a potential bias towards locality, which may happen when labeling is performed by exploring the graph starting from some node. For instance, in web search applications, a crawler has to explore the web graph following the links. Similarly, the information about the users of a social network can usually be obtained via an API, and new users are discovered following the friends of known users. To model such a situation, one could divide nodes based on the shortest path distance to a given node. However, graph distances are discrete, and the number of nodes at a certain distance may grow exponentially with distance. Thus, such an approach

does not provide us with the desired flexibility in varying the size of the train part. Instead, we use the concept of Personalized PageRank (PPR) [29] to define a local neighborhood of a node. PPR is the stationary distribution of a random walk that always restarts from some fixed node $j$. Thus, the personalization vector $\boldsymbol{p}$ in (1) is set to the one-hot-encoding of $j$. The associated distributional shift naturally captures locality since a random walk always restarts from the same node.

For our experiments, we select the node $j$ with the highest PR score as a restarting one. Then, we compute the PPR values $\pi_i$ for every node $i$ and define the measure $\sigma_i = -\pi_i$. The nodes with high PPR, which belong to the ID part, are expected to be close to the restarting node, while far away nodes go to the OOD subset. Figure 1b confirms that the locality is indeed preserved, as the ID part consists of one compact region around the restarting node. Thus, the OOD subset includes periphery nodes as well as some nodes that were previously marked as important in the PR-based split but are far away from the restarting node. Our analysis in Appendix A also provides evidence for this behavior: the PPR-based split strongly affects the distribution of pairwise distances within the ID/OOD parts as the locality bias of the ID part makes the OOD nodes more distant from each other.

While locality-based distributional shifts are natural, we are unaware of publicly available benchmarks focusing on such shifts. We believe that our approach will be helpful for evaluating the robustness of GNNs under such shifts. Our empirical results in Section 4 demonstrate that locality-based splits are the most challenging for graph models and thus may require special attention.

**Density-based**  The next distributional shift we propose is based on *density*. One of the most simple node characteristics that describe the local density in a graph is the local clustering coefficient. Considering some node $i$, let $d_i$ be its degree and $\gamma_i$ be the number of edges connecting the neighbors of $i$. Then, the local clustering coefficient is defined as the edge density within the one-hop neighborhood:

$$c_i = \frac{2\gamma_i}{d_i(d_i - 1)} \tag{2}$$

For our experiments, we consider nodes with the highest clustering coefficient to be ID, which implies that $\sigma_i = -c_i$.

This structural property might be particularly interesting for inducing distributional shifts since it is defined through the number of triangles, the substructures that most standard graph neural networks are unable to distinguish and count [4]. Thus, it is interesting to know how changes in the clustering coefficient affect the predictive performance and uncertainty estimation.

Figure 1c visualizes the density-based split. We see that the OOD part includes both the high-degree *central* nodes and the *periphery* nodes of degree one. Indeed, the nodes of degree one naturally have zero clustering coefficient. On the other hand, for the high-degree nodes, the number of edges between their neighbors usually grows slower than quadratically. Thus, such nodes tend to have a vanishing clustering coefficient.

Finally, we note that the existing datasets with realistic splits may often have the local clustering coefficient shifted between the train and test parts. Figures 12c and 13c in Appendix show this for two OGB datasets. Depending on the dataset, the train subset may be biased towards the nodes with either larger (Figure 13c) or smaller (Figure 12c) clustering. In our experiments, we focus on the former scenario.

## 4   Experimental setup

**Datasets**  While our approach can potentially be applied to any node prediction dataset, for our experiments, we pick the following seven homophilous datasets that are commonly used in the literature: three citation networks, including **CoraML**, **CiteSeer** [26, 9, 8, 31], and **PubMed** [28], two co-authorship graphs — **CoauthorPhysics** and **CoauthorCS** [32], and two co-purchase datasets — **AmazonPhoto** and **AmazonComputer** [25, 32]. Moreover, we consider **OGB-Products**, a large-scale dataset from the OGB benchmark. Some of the methods considered in our work are not able to process such a large dataset, so we provide only the analysis of structural shifts on this dataset and do not use it for comparing different methods.

For any distributional shift, we split each graph dataset as follows. The half of nodes with the smallest values of $\sigma_i$ are considered to be ID and split into `Train`, `Valid-In`, and `Test-In` uniformly at

random in proportion $30\% : 10\% : 10\%$. The second half contains the remaining OOD nodes and is split into `Valid-Out` and `Test-Out` in the ascending order of $\sigma_i$ in proportion $10\% : 40\%$. Thus, in our base setup, the ID to OOD split ratio is $50\% : 50\%$. We have also conducted experiments with other split ratios that involve smaller sizes of OOD subsets, see Appendix B for the details.

**Models**    In our experiments, we apply the proposed benchmark to evaluate the OOD robustness and uncertainty of various graph models. In particular, we consider the following methods for improving the OOD generalisation in the node classification task:

- **ERM** is the *Empirical Risk Minimization* technique that trains a simple GNN model by optimizing a standard classification loss;

- **DANN** is an instance of *Domain Adversarial Network* [7] that trains a regular and a domain classifiers to make features indistinguishable across different domains;

- **CORAL** is the *Deep Correlation Alignment* [34] technique that encourages the representations of nodes in different domains to be similar;

- **EERM** is the *Explore-to-Extrapolate Risk Minimization* [38] method based on graph structure editors that creates virtual environments during training;

- **Mixup** is an implementation of *Mixup* from [37] and represents a simple data augmentation technique adapted for the graph learning problems;

- **DE** represents a *Deep Ensemble* [21] of graph models, a strong but expensive method for improving the predictive performance;

In context of OOD detection, we consider the following uncertainty estimation methods:

- **SE** represents a simple GNN model that is used in the **ERM** method, for which the measure of uncertainty is *Softmax Entropy* (*i.e.*, the entropy of predictive distribution);

- **GPN** is an implementation of the *Graph Posterior Network* [33] method for the node-level uncertainty estimation;

- **NatPN** is an instance of *Natural Posterior Network* [3] in which the encoder has the same architecture as in the **SE** method;

- **DE** represents a *Deep Ensemble* [21] of graph models, which allows to separate the knowledge uncertainty that is used for OOD detection;

- **GPE** and **NatPE** represent the *Bayesian Combinations* of **GPN** and **NatPN**, an approach to construct an ensemble of Dirichlet models [3].

The training details are described in Appendix F. For experiments with the considered OOD robustness methods, including **DANN**, **CORAL**, **EERM**, and **Mixup**, we use the experimental framework from the GOOD benchmark,[4] whereas the remaining methods are implemented in our custom experimental framework and can be found in our repository.[5]

**Prediction tasks & evaluation metrics**    To evaluate OOD robustness in the node classification problem, we exploit standard *Accuracy*. Further, to assess the quality of uncertainty estimates, we treat the OOD detection problem as a binary classification with positive events corresponding to the observations from the OOD subset and use *AUROC* to measure performance.

## 5    Empirical study

In this section, we show how the proposed approach to creating distributional shifts can be used for evaluating the robustness and uncertainty of graph models. In particular, we compare the types of distributional shifts introduced above and discuss which of them are more challenging. We also discuss how they affect the predictive performance of OOD robustness methods as well as the ability for OOD detection of uncertainty estimation methods.

---

[4]Link to the GOOD benchmark repository
[5]Link to our GitHub repository

Table 1: Comparison of structural distributional shifts in terms of OOD robustness and OOD detection. We report the drop in predictive performance of the **ERM** method measured by *Accuracy* (left) and the quality of uncertainty estimates of the **SE** method measured by *AUROC* (right).

| | Popularity | Locality | Density | | Popularity | Locality | Density |
|---|---|---|---|---|---|---|---|
| AmazonComputer | $-10.01\%$ | $-21.95\%$ | $-7.91\%$ | AmazonComputer | 88.52 | 86.48 | 44.24 |
| AmazonPhoto | $-8.54\%$ | $-30.93\%$ | $-3.58\%$ | AmazonPhoto | 92.05 | 93.29 | 41.08 |
| CoauthorCS | $-3.13\%$ | $-1.22\%$ | $-4.86\%$ | CoauthorCS | 83.25 | 85.74 | 50.91 |
| CoauthorPhysics | $-3.42\%$ | $-4.75\%$ | $-1.41\%$ | CoauthorPhysics | 86.60 | 87.73 | 37.50 |
| CoraML | $-3.94\%$ | $-14.61\%$ | $-17.09\%$ | CoraML | 75.67 | 87.13 | 81.55 |
| CiteSeer | $-0.02\%$ | $-26.51\%$ | $-8.39\%$ | CiteSeer | 68.01 | 89.89 | 66.90 |
| PubMed | $-3.23\%$ | $-5.71\%$ | $-0.78\%$ | PubMed | 68.60 | 66.34 | 58.60 |
| OGB-Products | $-2.86\%$ | $-2.83\%$ | $-0.12\%$ | OGB-Products | 88.50 | 88.56 | 36.00 |
| Average | $-4.39\%$ | $-13.56\%$ | $-5.52\%$ | Average | 81.40 | 85.65 | 52.10 |

## 5.1 Analysis of structural distributional shifts

In this section, we analyze and compare the proposed structural distributional shifts.

**OOD robustness** To investigate how the proposed shifts affect the predictive performance of graph models, we take the most simple **ERM** method and report the drop in *Accuracy* between the ID and OOD test subsets in Table 1 (left). It can be seen that the node classification results on the considered datasets are consistently lower when measured on the OOD part, and this drop can reach tens of percent in some cases. The most significant decrease in performance is observed on the locality-based splits, where it reaches $14\%$ on average and more than $30\%$ in the worst case. This fact matches our intuition about how training on local regions of graphs may prevent OOD generalization and create a great challenge for improving OOD robustness. Although the density-based shift does not appear to be as difficult, it is still more challenging than the popularity-based shift, leading to performance drops of $5.5\%$ on average and $17\%$ in the worst case.

**OOD detection** To analyze the ability of graph models to detect distributional shifts by providing higher uncertainty on the shifted inputs, we report the performance of the most simple **SE** method for each proposed distributional shift and graph dataset in Table 1 (right). It can be seen that the popularity-based and locality-based shifts can be effectively detected by this method, which is proved by the average performance metrics. In particular, the *AUROC* values may vary from 68 to 92 points on the popularity-based splits and approximately in the same range for the locality-based splits. Regarding the density-based shifts, one can see that the OOD detection performance is almost the same as for random predictions on average. Only for the citation networks the *AUROC* exceeds 58 points, reaching a peak of 81 points. This is consistent with the previous works showing that standard graph neural networks are unable to count substructures such as triangles. In our case, this leads to graph models failing to detect changes in density measured as the number of triangles around the central node.

Thus, the popularity-based shift appears to be the simplest for OOD detection, while the density-based is the most difficult. This clearly shows how our approach to creating data splits allows one to vary the complexity of distributional shifts using different structural properties as splitting factors.

## 5.2 Comparison of existing methods

In this section, we compare several existing methods for improving OOD robustness and detecting OOD inputs on the proposed structural shifts. To concisely illustrate the overall performance of the models, we first rank them according to a given performance measure on a particular dataset and then average the results over the datasets. For detailed results of experiments on each graph dataset separately, please refer to Appendix G.

**OOD robustness** For each model, we measure both the absolute values of *Accuracy* and the drop in this metric between the ID and OOD subsets in Table 2 (left). It can be seen that a simple data augmentation technique **Mixup** often shows the best performance. Only on the density-based shift, expensive **DE** outperforms it on average when tested on the OOD subset. This proves that data

Table 2: Comparison of several graph methods for improving the OOD robustness (left) and detecting the OOD inputs by means of uncertainty estimation (right). For each task, we report the method ranks averaged across different graph datasets (lower is better).

| | Popularity | | Locality | | Density | | | | Popularity | Locality | Density |
|---|---|---|---|---|---|---|---|---|---|---|---|
| | ID | OOD | ID | OOD | ID | OOD | | | | | |
| ERM | 4.0 | 4.0 | 3.3 | 4.1 | 3.9 | 4.0 | | SE | 1.4 | 2.1 | 4.0 |
| Mixup | 1.4 | 2.1 | 1.4 | 2.4 | 1.9 | 3.3 | | GPN | 3.3 | 3.9 | 4.3 |
| EERM | 3.6 | 3.9 | 4.4 | 3.3 | 5.0 | 4.3 | | NatPN | 5.3 | 4.1 | 2.9 |
| DANN | 4.3 | 4.3 | 5.0 | 4.1 | 3.0 | 3.6 | | DE | 2.1 | 1.1 | 2.7 |
| CORAL | 4.7 | 4.1 | 4.3 | 4.7 | 4.1 | 3.9 | | GPE | 3.1 | 4.3 | 3.4 |
| DE | 3.0 | 2.6 | 2.6 | 2.3 | 3.1 | 2.0 | | NatPE | 5.7 | 5.4 | 3.7 |

Table 3: Comparison of the proposed architecture modifications that are used in the **ERM** method for OOD robustness (left) and **SE** method for OOD detection (right). For each task, we report the `win/tie/loss` counts across graph datasets for the modified GNN architecture against the base one.

| | Popularity | | Locality | | Density | | | | Popularity | Locality | Density |
|---|---|---|---|---|---|---|---|---|---|---|---|
| | ID | OOD | ID | OOD | ID | OOD | | | | | |
| ERM + mod | 4/2/1 | 5/2/0 | 4/2/1 | 3/2/2 | 4/2/1 | 5/2/0 | | SE + mod | 0/0/7 | 0/1/6 | 4/0/3 |

augmentation techniques are beneficial in practice, as they prevent overfitting to the ID structural patterns and improve the OOD robustness. Regarding other OOD robustness methods, a graph-specific method **EERM** outperforms **DANN** and **CORAL** on the popularity-based and locality-based shifts. However, these domain adaptation methods are superior to **EERM** on the density-based shifts, providing better predictive performance on average for both ID and OOD subsets.

In conclusion, we reveal that the most sophisticated methods that generate virtual environments and predict the underlying domains for improving the OOD robustness may often be outperformed by simpler methods, such as data augmentation.

**OOD detection**  Comparing the quality of uncertainty estimates in Table 2 (right), one can observe that the methods based on the entropy of predictive distribution usually outperform Dirichlet methods. In particular, a natural distinction of knowledge uncertainty in **DE** enables it to produce uncertainty estimates that are the most consistent with OOD inputs on average, especially when applied to the locality-based and density-based shifts. In general, **GPN** and the combination of its instances **GPE** provide higher OOD detection performance than their counterparts based on **NatPN** when tested on the popularity-based and locality-based splits.

### 5.3  Influence of graph architecture improvements

In this section, we consider several adjustments for the base GNN architecture of such methods as **ERM**, which is used to evaluate OOD robustness, and **SE**, which provides the uncertainty estimates for OOD detection. In particular, we reduce the number of graph convolutional layers from 3 to 2, replacing the first one with a pre-processing step based on MLP, apply the skip-connections between graph convolutional layers, and replace the GCN [16] graph convolution with SAGE [11].

These changes are aimed at relaxing the restrictions on the information exchange between a central node and its neighbors and providing more independence in processing the node representations across neural layers. Such a modification is expected to help the GNN model to learn structural patterns that could be transferred to the shifted OOD subset more successfully. Further, we investigate how these changes in the model architecture affect the predictive performance and the quality of uncertainty estimates of the corresponding methods when tested on the proposed structural distributional shifts.

For this, we use the `win/tie/loss` counts that reflect how many times the modified architecture has outperformed, got a statistically insignificant difference, or lost to the corresponding method, respectively. As can be seen from Table 3 (left), the **ERM** method supplied with the proposed modifications usually outperforms the baseline architecture both on ID and OOD, which is proved by high `win` counts. However, as can be seen from Table 3 (right), the same modification in the corresponding **SE** method leads to consistent performance drops, which is reflected in high `loss`

counts. It means that, when higher predictive performance is reached on the shifted subset, it becomes more difficult to detect the inputs from this subset as OOD since they appear to be less distinguishable by a GNN model in the context of the base node classification problem. Our observation is very similar to what is required from *Invariant Learning* techniques, which try to produce node representations invariant to different domains or environments. This may serve as evidence that there is a trade-off between the quality of learned representations for solving the target node classification task under structural distributional shift and the ability to separate the nodes from different distributions based on these representations.

## 6 Conclusion

In this work, we propose and analyze structural distributional shifts for evaluating robustness and uncertainty in node-level graph problems. Our approach allows one to create realistic, challenging, and diverse distributional shifts for an arbitrary graph dataset. In our experiments, we evaluate the proposed structural shifts and show that they can be quite challenging for existing graph models. We also find that simple models often outperform more sophisticated methods on these challenging shifts. Moreover, by applying various modifications for graph model architectures, we show that there is a trade-off between the quality of learned representations for the target classification task under structural distributional shift and the ability to detect the shift using these representations.

**Limitations**   While our methods of creating structural shifts are motivated by real distributional shifts that arise in practice, they are synthetically generated, whereas, for particular applications, natural distributional shifts would be preferable. However, our goal is to address the situations when such natural shifts are unavailable. Thus, we have chosen an approach universally applied to any dataset. Importantly, graph structure is the only common modality of different graph datasets that can be exploited in the same manner to model diverse and complex distributional shifts.

**Broader impact**   Considering the broader implications of our work, we assume that the proposed approach for evaluating robustness and uncertainty of graph models will support the development of more reliable systems based on machine learning. By testing on the presented structural shifts, it should be easier to detect various biases against under-represented groups that may have a negative impact on the resulting performance and interfere with fair decision-making.

**Future work**   In the future, two key areas can be explored based on our work. Firstly, there is a need to develop principled solutions for improving robustness and uncertainty estimation on the proposed structural shifts. Our new approach can assist in achieving this objective by providing an instrument for testing and evaluating such solutions. Additionally, there is a need to create new graph benchmarks that accurately reflect the properties of data observed in real-world applications. This should involve replacing synthetic shifts with realistic ones. By doing so, we may capture the challenges and complexities that might be faced in practice, thereby enabling the development of more effective and applicable graph models.

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

# A Properties of distributional shifts

This section provides a detailed analysis and comparison of the proposed distributional shifts. For this purpose, we consider three representative real-world datasets **AmazonComputer**, **CoauthorCS**, and **CoraML**, and discuss how different distributional shifts affect the basic properties of data: *class balance*, *degree distribution*, and *graph distances* between nodes within ID and OOD subsets.

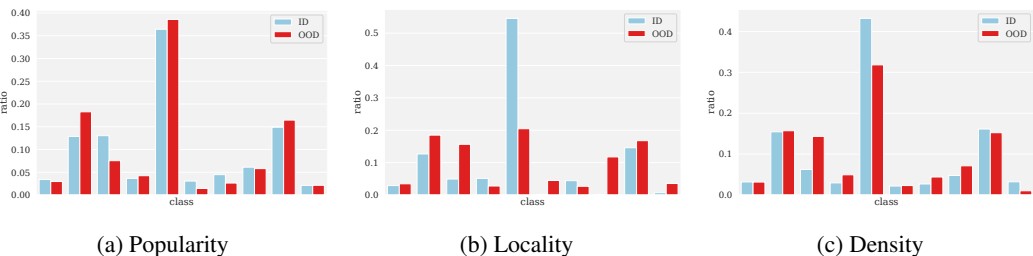

| (a) Popularity | (b) Locality | (c) Density |

Figure 2: Class balance for **AmazonComputer** dataset across different types of shifts.

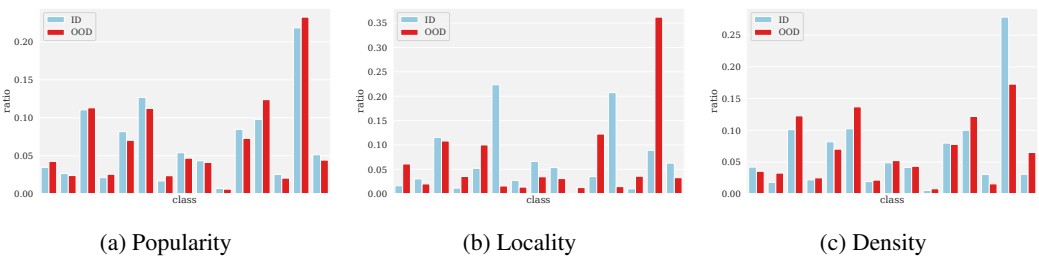

| (a) Popularity | (b) Locality | (c) Density |

Figure 3: Class balance for **CoauthorCS** dataset across different types of shifts.

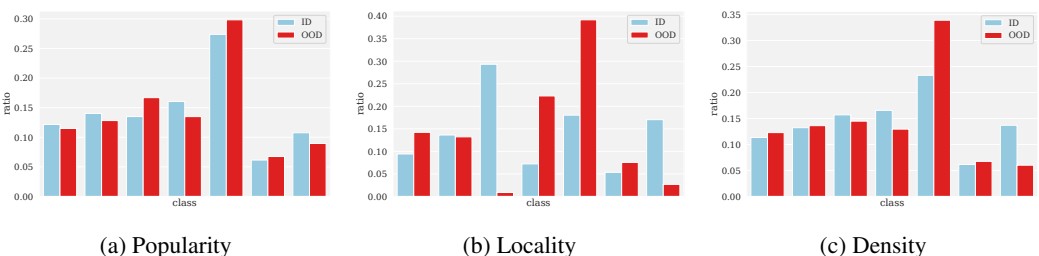

| (a) Popularity | (b) Locality | (c) Density |

Figure 4: Class balance for **CoraML** dataset across different types of shifts.

**Class balance**    Class balance directly affects the amount of evidence acquired by the graph processing model and used for estimating uncertainty and making predictions. It is especially important for evaluating Dirichlet-based models which exploit normalizing flows, as their density estimates can become irrelevant due to a significant change in class balance.

In Figures 2–4, one can see that the popularity-based split does not create a notable difference in the class balance between the ID and OOD subsets (for the datasets under consideration). Thus, the more important and less important nodes have, on average, the same probability of belonging to a particular class. More noticeable differences are induced by the density-based split. The locality-based split leads to the most significant changes for some classes. This shows that the split strategies based on the structural locality in graph can be very challenging as they also affect such crucial statistics as class balance.

**Degree distribution**    The node degree distribution is one of the basic structural characteristics of graph that describes the local importance of nodes. Degrees are especially important for such graph

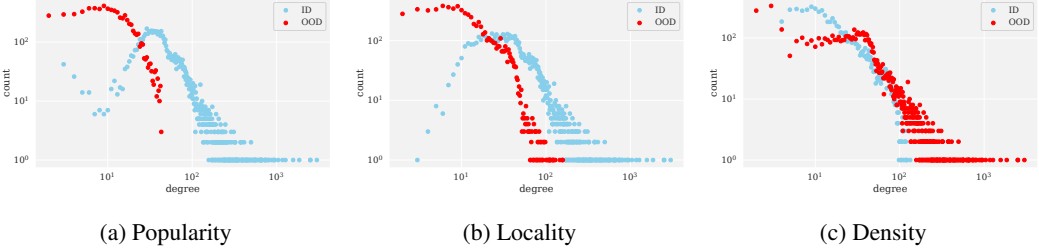

(a) Popularity            (b) Locality            (c) Density

Figure 5: The distribution of node degrees for **AmazonComputer** dataset across different shifts.

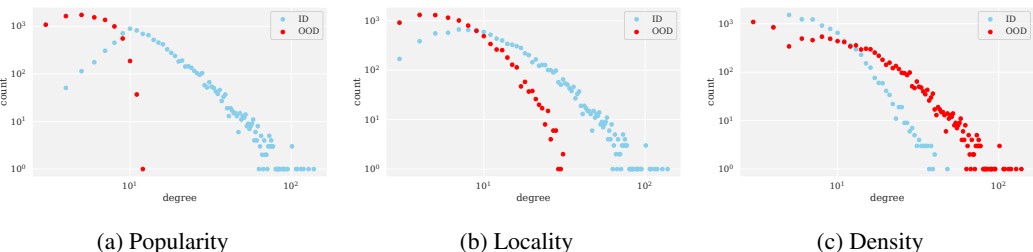

(a) Popularity            (b) Locality            (c) Density

Figure 6: The distribution of node degrees for **CoauthorCS** dataset across different shifts.

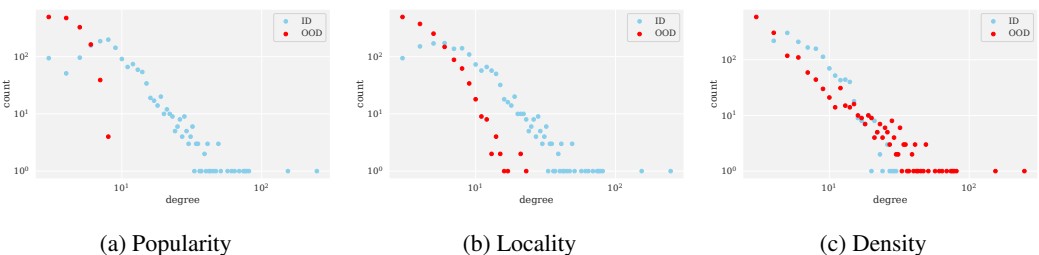

(a) Popularity            (b) Locality            (c) Density

Figure 7: The distribution of node degrees for **CoraML** dataset across different shifts.

processing methods as GNNs since they describe how many channels around the considered node are used for message passing and aggregation.

In Figures 5–7, one can see that the most significant change in the degree distribution appears when the ID and OOD subsets are separated based on PageRank: the ID part contains more high-degree nodes. This is expected since PageRank is a graph characteristic measuring node importance (*a.k.a.* centrality), and node degree is the simplest centrality measure known to be correlated with PageRank. For the locality-based splits, the difference in degree distribution is smaller but still significant since PPR selects nodes by their relative importance for a particular node, so some high-degree nodes can be less important. Finally, for the density-based splits, the degree distribution also changes, but the extent of shift is usually less significant, and for some datasets (*e.g.*, **CoauthorCS**), the higher-degree nodes are in the OOD subset.

**Graph distance distribution** The distance between two nodes in a graph is defined as the length of the shortest path between them. Here, we compute such distances between the nodes in the ID or OOD subset within the original graph, *i.e.*, we consider the whole graph when searching for the shortest path. The distribution of distances illustrates which parts of the datasets are more locally concentrated.

In Figures 8–10, one can observe that the locality-based split leads to the most significant changes in distances, making the OOD nodes nearly twice as far from each other as the ID ones. At the same time, the popularity-based split does not lead to such a difference, revealing almost the same distributions on ID and OOD subsets. This means that the popularity bias in a graph does not prevent one from

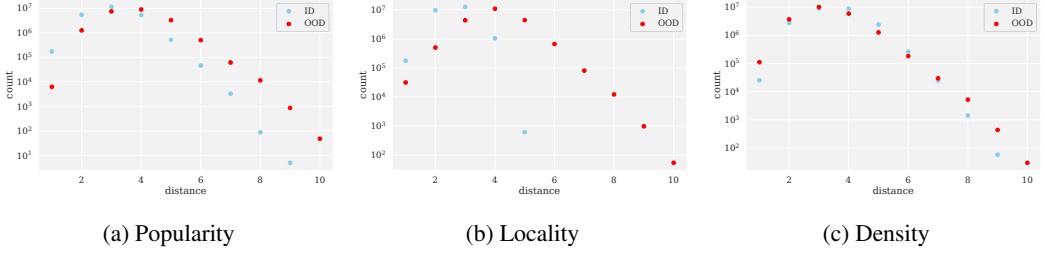

(a) Popularity       (b) Locality       (c) Density

Figure 8: The distribution of graph distances for **AmazonComputer** dataset across different shifts.

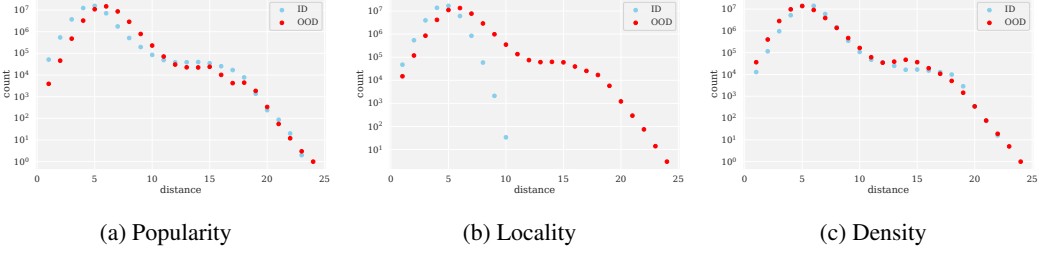

(a) Popularity       (b) Locality       (c) Density

Figure 9: The distribution of graph distances for **CoauthorCS** dataset across different shifts.

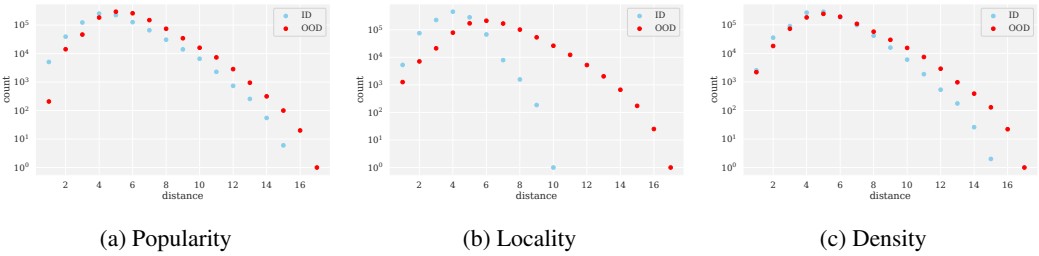

(a) Popularity       (b) Locality       (c) Density

Figure 10: The distribution of graph distances for **CoraML** dataset across different shifts.

covering the less popular *periphery* nodes since the most popular nodes may be widespread. Similarly, the density-based split does not induce a significant difference in pairwise distances.

# B    Results on other split ratios

The main part of our empirical study is focused on a $50\% : 50\%$ split ratio. In this section, we extend our discussion with varying ID to OOD ratios and consider two setups where the fraction of OOD samples is smaller — $70\% : 30\%$ and $90\% : 10\%$.

Let us first revisit our analysis of the proposed structural shifts on a $70\% : 30\%$ ratio. As can be seen in Table 4, the results in both OOD robustness, which is evaluated in the node classification task, and OOD detection, which is done by means of uncertainty estimation, have not changed much compared to our base setup discussed in the main text — the considered graph models show the average drop in performance of $3\%$ on the popularity-based shift and $6\%$ on the density-based shift, while the OOD detection performance remains at nearly $81$ and $54$ points, respectively. Despite the fact that the graph models now have access to more diverse structures in terms of popularity and local density, making decisions about unimportant and sparsely surrounded nodes remains as difficult as before.

Further, the drop in predictive performance on the locality-based shift appears to be less significant compared to the original setup and reaches only $5\%$ on average instead of the previous $14\%$. At the same time, the OOD detection performance on this structural shift drops to $79$ points, whereas it was $85$ points in the original setup. These results also match our intuition — as the distance between the ID and OOD nodes decreases, the OOD samples become less distinguishable from the ID ones. This makes the OOD detection performance drop on average, while the gap between ID and OOD metrics in standard node classification disappears.

We also may compare the existing graph models based on the results in Table 5. As can be seen, the ranking of methods remains almost the same as in the original setup. Specifically, the most simple data augmentation technique **Mixup** often shows the best performance in OOD robustness, while **DE** provides the second best results on most structural shifts. As for OOD detection, the methods based on the entropy of predictive distribution again outperform the Dirichlet ones, and the uncertainty estimates produced by **DE** are best correlated with the OOD examples. Thus, our observations regarding the performance of graph models are consistent with those reported in the paper.

Table 4: Comparison of structural distributional shifts in terms of OOD robustness and OOD detection. We report the drop in predictive performance of the **ERM** method measured by *Accuracy* (left) and the quality of uncertainty estimates of the **SE** method measured by *AUROC* (right). The ID to OOD ratio in these experiments is $70\% : 30\%$ instead of $50\% : 50\%$ as in the main text.

| | Popularity | Locality | Density | | Popularity | Locality | Density |
|---|---|---|---|---|---|---|---|
| AmazonComputer | $-3.80\,\%$ | $-12.66\,\%$ | $-14.04\,\%$ | AmazonComputer | 87.83 | 85.77 | 50.46 |
| AmazonPhoto | $-7.19\,\%$ | $-3.03\,\%$ | $-6.79\,\%$ | AmazonPhoto | 91.80 | 85.78 | 47.27 |
| CoauthorCS | $-6.51\,\%$ | $-1.89\,\%$ | $-6.64\,\%$ | CiteSeer | 70.56 | 64.55 | 58.04 |
| CoauthorPhysics | $-2.58\,\%$ | $-5.50\,\%$ | $-2.47\,\%$ | CoauthorCS | 83.68 | 81.13 | 63.24 |
| CoraML | $-7.56\,\%$ | $-13.63\,\%$ | $-9.49\,\%$ | CoauthorPhysics | 86.70 | 82.06 | 39.68 |
| CiteSeer | $+3.24\,\%$ | $+2.50\,\%$ | $-4.13\,\%$ | CoraML | 82.06 | 84.63 | 76.82 |
| PubMed | $+0.46\,\%$ | $-0.67\,\%$ | $-1.76\,\%$ | PubMed | 66.64 | 63.34 | 55.79 |
| OGB-Products | $-2.35\,\%$ | $-2.36\,\%$ | $-4.02\,\%$ | OGB-Products | 82.35 | 82.44 | 43.50 |
| Average | $-3.28\,\%$ | $-4.65\,\%$ | $-6.17\,\%$ | Average | 81.45 | 78.71 | 54.35 |

Table 5: Comparison of several graph methods for improving the OOD robustness (left) and detecting the OOD inputs by means of uncertainty estimation (right). For each task, we report the method ranks averaged across different graph datasets (lower is better). The ID to OOD ratio in these experiments is $70\% : 30\%$ instead of $50\% : 50\%$ as in the main text.

| | Popularity | | Locality | | Density | | | Popularity | Locality | Density |
|---|---|---|---|---|---|---|---|---|---|---|
| | ID | OOD | ID | OOD | ID | OOD | | | | |
| ERM | 3.0 | 3.4 | 2.9 | 3.1 | 4.3 | 4.3 | SE | 1.3 | 2.3 | 3.7 |
| Mixup | 1.9 | 2.6 | 1.4 | 2.7 | 1.4 | 3.3 | GPN | 3.3 | 3.9 | 3.9 |
| EERM | 4.9 | 3.9 | 5.0 | 4.0 | 4.4 | 4.1 | NatPN | 5.4 | 4.3 | 3.7 |
| DANN | 4.1 | 4.4 | 4.3 | 4.3 | 3.7 | 2.6 | DE | 3.0 | 1.4 | 1.9 |
| CORAL | 4.1 | 4.0 | 4.4 | 4.9 | 4.1 | 3.7 | GPE | 3.1 | 3.9 | 3.4 |
| DE | 3.0 | 2.7 | 3.0 | 2.0 | 3.0 | 3.0 | NatPE | 4.9 | 5.3 | 4.4 |

Now let us discuss the setup based on a more extreme $90\% : 10\%$ ratio. According to Table 6, the drop in predictive performance under the popularity-based and locality-based shifts reaches more than $6\%$ instead of $3\%$ in the previous setup and almost $11\%$ instead of $6\%$, respectively. This was expected and can be explained by the fact that graph models are now tested on the nodes with the most extreme structural properties (i.e., nodes with the lowest PageRank or clustering coefficient), and their inaccurate predictions are no longer compensated by the more accurate ones, which were made on the nodes with far less anomalous properties.

At the same time, the performance drop on the locality-based shift appears to be nearly $1\%$ on average. This result is quite reasonable since now graph models have access to the whole variety of graph substructures and node features, which allows them to predict equally well for any graph region regardless of its distance to some particular node. The observed effects are interesting and also important to consider when using our approach for evaluating graph models.

Regarding the OOD detection performance, we observe that results on the locality-based shift keep decreasing and reach 73 points, in contrast to 79 points in the previous setup (and this might happen for the same reason as we discussed before). At the same time, the detection metrics on the remaining structural shifts appear to be nearly the same (79 points instead of the previous 81 on the popularity-based shift) or even better on average (63 points instead of 54 on the density-based shift), which is consistent with the changes in predictive performance discussed above.

As for the ranking of different models in Table 7, it remains almost the same, with the most simple methods providing top performance on the majority of prediction tasks.

Table 6: Comparison of structural distributional shifts in terms of OOD robustness and OOD detection. We report the drop in predictive performance of the **ERM** method measured by *Accuracy* (left) and the quality of uncertainty estimates of the **SE** method measured by *AUROC* (right). The ID to OOD ratio in these experiments is $90\% : 10\%$ instead of $50\% : 50\%$ as in the main text.

| | Popularity | Locality | Density | | Popularity | Locality | Density |
|---|---|---|---|---|---|---|---|
| AmazonComputer | $-8.45\%$ | $-7.79\%$ | $-30.98\%$ | AmazonComputer | 83.42 | 76.53 | 68.97 |
| AmazonPhoto | $-8.00\%$ | $+0.12\%$ | $-16.35\%$ | AmazonPhoto | 89.01 | 82.57 | 64.41 |
| CoauthorCS | $-9.79\%$ | $-4.42\%$ | $-7.76\%$ | CiteSeer | 75.90 | 65.11 | 55.48 |
| CoauthorPhysics | $-3.77\%$ | $-4.91\%$ | $-5.32\%$ | CoauthorCS | 83.22 | 76.02 | 70.63 |
| CoraML | $-16.10\%$ | $+5.52\%$ | $-8.03\%$ | CoauthorPhysics | 84.68 | 78.60 | 52.97 |
| CiteSeer | $-5.14\%$ | $+1.35\%$ | $+1.86\%$ | CoraML | 79.36 | 69.64 | 69.78 |
| PubMed | $+3.33\%$ | $+4.70\%$ | $-2.63\%$ | PubMed | 61.05 | 57.14 | 55.11 |
| OGB-Products | $-2.92\%$ | $-2.75\%$ | $-15.61\%$ | OGB-Products | 77.92 | 78.02 | 68.33 |
| Average | $-6.36\%$ | $-1.02\%$ | $-10.60\%$ | Average | 79.32 | 72.95 | 63.21 |

Table 7: Comparison of several graph methods for improving the OOD robustness (left) and detecting the OOD inputs by means of uncertainty estimation (right). For each task, we report the method ranks averaged across different graph datasets (lower is better). The ID to OOD ratio in these experiments is $90\% : 10\%$ instead of $50\% : 50\%$ as in the main text.

| | Popularity | | Locality | | Density | | | Popularity | Locality | Density |
|---|---|---|---|---|---|---|---|---|---|---|
| | ID | OOD | ID | OOD | ID | OOD | | | | |
| ERM | 3.3 | 3.7 | 3.4 | 3.0 | 3.3 | 4.4 | SE | 1.6 | 2.6 | 3.3 |
| Mixup | 1.7 | 2.4 | 1.7 | 2.9 | 1.7 | 3.0 | GPN | 3.1 | 3.0 | 3.9 |
| EERM | 5.0 | 4.1 | 5.0 | 3.7 | 4.9 | 3.9 | NatPN | 5.0 | 4.4 | 3.7 |
| DANN | 4.3 | 4.0 | 4.0 | 4.3 | 3.4 | 3.0 | DE | 2.6 | 1.7 | 1.6 |
| CORAL | 3.9 | 3.6 | 4.1 | 5.1 | 4.3 | 3.4 | GPE | 3.6 | 4.0 | 4.3 |
| DE | 2.9 | 3.1 | 2.7 | 2.0 | 3.4 | 3.3 | NatPE | 5.1 | 5.3 | 4.3 |

# C Comparison with the GOOD benchmark from Gui et al. [10]

Our work complements and extends the GOOD benchmark recently proposed by Gui et al. [10]. However, there are several important differences that we discuss in this section.

One of the main properties of the GOOD benchmark is its theoretical distinction between two types of distributional shifts, which are represented through a graphical model. In particular, the authors consider covariate shifts, in which the distribution of features changes while the conditional distribution of targets given features remains the same, and concept shifts, where the opposite situation occurs, *i.e.*, the conditional target distribution changes, while the feature distribution is the same. Although this distinction might be very helpful for understanding the properties of particular GNN models, such exclusively covariate or concept shifts rarely happen in practice where both types of shifts are present at the same time.

To create pure covariate or concept shifts, Gui et al. [10] introduce different subsets of variables that either fully determine the target, create confounding associations with the target, or are completely independent of the target. This has to be properly handled and makes it non-trivial to create distributional shifts on new datasets with this approach. Indeed, the distributional shifts in the GOOD benchmark can be properly implemented only for synthetic graph datasets or via appending synthetic features that either describe various domains as completely independent variables or create the necessary concepts by inducing some spurious correlation with the target. Moreover, the authors claim that, in the case of real-world datasets, one has to perform screening over the available node features to create the required setup of domain or concept shift. This fact implies numerous restrictions on how the data splits can be prepared.

In contrast, our method does not distinguish between covariate and concept shifts and thus can be universally applied to any dataset and does not require any dataset modifications. Importantly, the type of distributional shift and the sizes of all split parts are easily controllable. This flexibility is the main advantage of our approach.

Finally, Gui et al. [10] confirm the importance of using both node features and graph structure. Still, their node-level distributional shifts are mainly based on node features such as the number of words or the year of publication in a citation network, the language of users in a social network, or the name of organizations in a webpage network. As for the graph properties, only node degrees are used in some citation networks. In contrast, we focus on the graph structure and propose diverse structural shifts together with a framework allowing one to easily create splits based on other structural properties.

# D Structural properties of the OGB data splits proposed by Hu et al. [12]

In Figures 11–13, we present how the distribution of structural node characteristics, including PageRank, Personalized PageRank, and clustering coefficient, may change across the train and test parts in some OGB datasets, where the distributional shifts are constructed using some domain-specific node feature. This shows that our approach to creating data splits using structural graph properties can be similar to realistic distributional shifts.

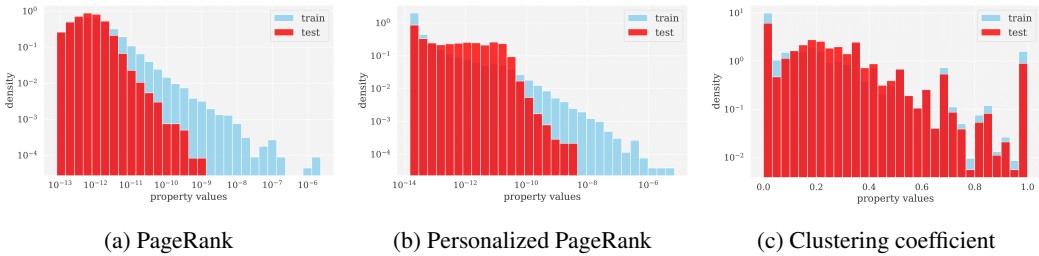

    (a) PageRank        (b) Personalized PageRank     (c) Clustering coefficient

Figure 11: Structural properties across train and test parts in *time-based* split of **OGB-Arxiv**.

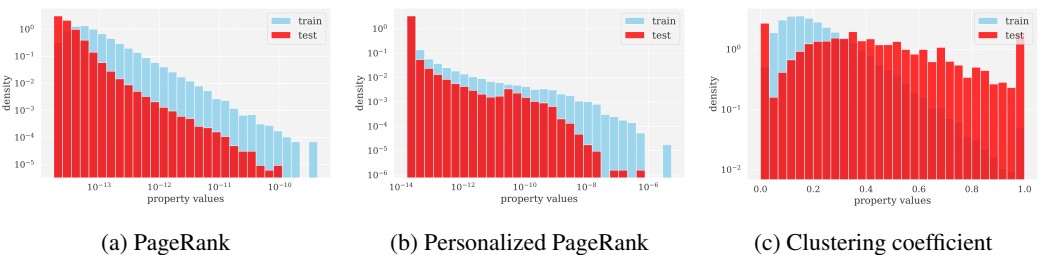

    (a) PageRank        (b) Personalized PageRank     (c) Clustering coefficient

Figure 12: Structural properties across train and test parts in *rank-based* split of **OGB-Products**.

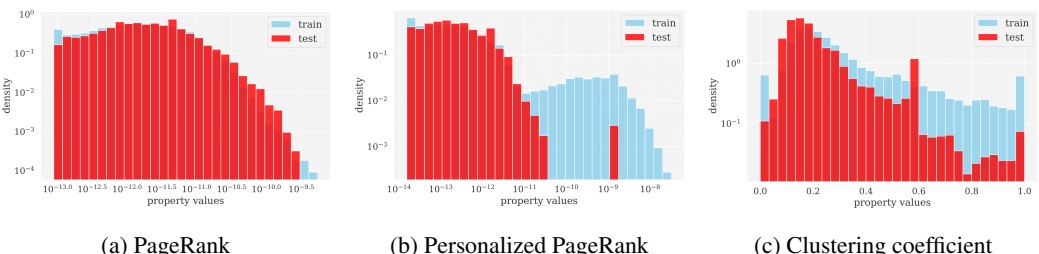

    (a) PageRank        (b) Personalized PageRank     (c) Clustering coefficient

Figure 13: Structural properties across train and test parts in *species-based* split of **OGB-Proteins**.

# E Dataset characteristics

Characteristics of the datasets used in this paper are listed in Table 8.

Table 8: Description of the considered graph datasets for node classification task.

| Dataset | # Nodes | # Edges | # Classes | # Features |
|---|---|---|---|---|
| AmazonComputer | 13,381 | 259,159 | 10 | 767 |
| AmazonPhoto | 7,484 | 126,530 | 8 | 745 |
| CoauthorCS | 18,333 | 163,788 | 15 | 6,805 |
| CoauthorPhysics | 34,493 | 495,924 | 5 | 8,415 |
| CoraML | 2,995 | 16,316 | 7 | 2,879 |
| CiteSeer | 3,327 | 4,732 | 6 | 3,703 |
| PubMed | 19,717 | 44,338 | 3 | 8,415 |
| OGB-Products | 2,449,029 | 61,859,140 | 47 | 100 |

# F Method configurations

For our main series of experiments with graph models in Sections 5.1 and 5.2, we consider a graph encoder based on three GCN convolutional layers [16]. On top of this graph encoder, we use a single linear layer as the task-specific head, which is used to predict the parameters of categorical distribution in classification tasks, the parameters of Dirichlet distribution for modeling uncertainty, *etc*. The hidden dimension of our baseline architecture is 256, and the dropout between hidden layers is $p = 0.2$. We exploit a standard Adam optimizer [14] with a learning rate of 0.0003 and a weight decay of 0.00001. For an additional series of experiments with an improved GNN architecture, which is discussed in Section 5.3, we reduce the number of graph convolutional layers from 3 to 2, replacing the first one with a pre-processing step based on linear layer, apply the skip-connections between graph convolutional layers, and replace the GCN graph convolution with SAGE [11].

Some of the considered methods for improving the OOD robustness, such as **DANN** and **CORAL**, exploit the notion of domains and require the knowledge about which nodes in the training set belong to which domain. Therefore, we need to define domains, which is not straightforward since our node properties are real-valued, not discrete. Thus, we discretize the values into $k = 10$ non-intersecting domains. After that, we assign a domain index to each node. These indices are treated as labels and used by **DANN** and **CORAL** during their training.

# G  Detailed experimental results

In this section, we provide detailed experimental results.

- Tables 9 — 15 show the predictive performance of various OOD robustness methods on the proposed structural distributional shifts;
- Tables 16 — 22 show the OOD detection performance of various uncertainty estimation methods on the proposed structural distributional shifts.

Table 9: Comparison of several graph methods for improving the OOD robustness in terms of their predictive performance across structural shifts on **CoraML** dataset.

|           | popularity |           | locality  |           | density   |           |
|-----------|------------|-----------|-----------|-----------|-----------|-----------|
|           | ID         | OOD       | ID        | OOD       | ID        | OOD       |
| ERM       | $85.80 \pm 1.10$ | $82.42 \pm 0.79$ | $84.47 \pm 0.69$ | $72.13 \pm 1.74$ | $93.27 \pm 0.28$ | $77.33 \pm 1.07$ |
| Mixup     | $89.27 \pm 0.57$ | $83.65 \pm 0.67$ | $86.27 \pm 0.57$ | $78.42 \pm 0.84$ | $88.20 \pm 0.44$ | $78.98 \pm 2.10$ |
| EERM      | $88.40 \pm 1.22$ | $83.07 \pm 0.76$ | $88.80 \pm 0.73$ | $69.96 \pm 6.62$ | $86.87 \pm 0.45$ | $79.07 \pm 0.32$ |
| DANN      | $86.00 \pm 1.10$ | $79.01 \pm 0.77$ | $81.45 \pm 0.65$ | $65.03 \pm 2.51$ | $91.33 \pm 0.76$ | $78.37 \pm 2.07$ |
| CORAL     | $85.00 \pm 1.11$ | $77.65 \pm 0.62$ | $82.33 \pm 0.25$ | $66.83 \pm 2.24$ | $89.55 \pm 0.54$ | $73.34 \pm 2.28$ |
| DE        | $87.00 \pm 0.00$ | $82.74 \pm 0.00$ | $84.67 \pm 0.00$ | $75.40 \pm 0.00$ | $93.00 \pm 0.00$ | $78.90 \pm 0.00$ |
| ERM + mod | $85.33 \pm 0.62$ | $82.55 \pm 0.56$ | $84.87 \pm 0.61$ | $72.59 \pm 0.96$ | $94.13 \pm 0.69$ | $79.33 \pm 0.37$ |

Table 10: Comparison of several graph methods for improving the OOD robustness in terms of their predictive performance across structural shifts on **CiteSeer** dataset.

|           | popularity |           | locality  |           | density   |           |
|-----------|------------|-----------|-----------|-----------|-----------|-----------|
|           | ID         | OOD       | ID        | OOD       | ID        | OOD       |
| ERM       | $72.43 \pm 1.33$ | $72.42 \pm 0.37$ | $77.60 \pm 0.66$ | $57.03 \pm 1.16$ | $73.75 \pm 0.96$ | $67.57 \pm 0.49$ |
| Mixup     | $72.79 \pm 0.74$ | $72.13 \pm 0.69$ | $76.82 \pm 0.59$ | $56.45 \pm 4.53$ | $76.88 \pm 1.47$ | $68.59 \pm 2.12$ |
| EERM      | $71.47 \pm 1.01$ | $70.48 \pm 0.66$ | $75.31 \pm 0.80$ | $61.95 \pm 3.30$ | $76.46 \pm 0.92$ | $68.18 \pm 0.71$ |
| DANN      | $67.57 \pm 0.96$ | $67.84 \pm 0.65$ | $71.17 \pm 0.33$ | $54.90 \pm 2.49$ | $75.07 \pm 1.48$ | $61.13 \pm 2.82$ |
| CORAL     | $67.87 \pm 1.19$ | $67.77 \pm 0.38$ | $71.87 \pm 0.62$ | $57.16 \pm 1.84$ | $73.47 \pm 1.22$ | $58.53 \pm 2.02$ |
| DE        | $73.27 \pm 0.00$ | $72.37 \pm 0.00$ | $78.38 \pm 0.00$ | $64.71 \pm 0.00$ | $74.17 \pm 0.00$ | $70.35 \pm 0.00$ |
| ERM + mod | $73.75 \pm 0.62$ | $71.73 \pm 1.02$ | $76.70 \pm 1.14$ | $59.86 \pm 2.23$ | $75.80 \pm 0.96$ | $68.32 \pm 1.01$ |

Table 11: Comparison of several graph methods for improving the OOD robustness in terms of their predictive performance across structural shifts on **PubMed** dataset.

|           | popularity |           | locality  |           | density   |           |
|-----------|------------|-----------|-----------|-----------|-----------|-----------|
|           | ID         | OOD       | ID        | OOD       | ID        | OOD       |
| ERM       | $86.85 \pm 0.12$ | $84.04 \pm 0.33$ | $86.75 \pm 0.48$ | $81.80 \pm 1.07$ | $85.96 \pm 0.27$ | $85.29 \pm 0.16$ |
| Mixup     | $89.32 \pm 0.24$ | $88.21 \pm 0.21$ | $87.94 \pm 0.37$ | $86.20 \pm 0.75$ | $88.72 \pm 0.25$ | $88.23 \pm 0.16$ |
| EERM      | $86.83 \pm 0.12$ | $83.39 \pm 0.10$ | $85.67 \pm 0.24$ | $84.79 \pm 0.18$ | $86.43 \pm 0.22$ | $84.10 \pm 0.10$ |
| DANN      | $86.26 \pm 0.48$ | $84.49 \pm 0.18$ | $86.04 \pm 0.30$ | $85.41 \pm 1.04$ | $87.07 \pm 0.24$ | $84.74 \pm 0.14$ |
| CORAL     | $86.31 \pm 0.48$ | $84.48 \pm 0.37$ | $86.00 \pm 0.32$ | $85.36 \pm 1.63$ | $87.24 \pm 0.15$ | $84.86 \pm 0.27$ |
| DE        | $87.17 \pm 0.00$ | $84.72 \pm 0.00$ | $87.07 \pm 0.00$ | $82.82 \pm 0.00$ | $86.61 \pm 0.00$ | $85.97 \pm 0.00$ |
| ERM + mod | $88.83 \pm 0.24$ | $87.39 \pm 0.27$ | $88.32 \pm 0.51$ | $87.58 \pm 0.17$ | $88.56 \pm 0.43$ | $87.71 \pm 0.22$ |

Table 12: Comparison of several graph methods for improving the OOD robustness in terms of their predictive performance across structural shifts on **AmazonComputer** dataset.

| | popularity | | locality | | density | |
|---|---|---|---|---|---|---|
| | ID | OOD | ID | OOD | ID | OOD |
| ERM | $92.43 \pm 0.24$ | $83.18 \pm 0.58$ | $93.04 \pm 0.38$ | $72.62 \pm 0.52$ | $93.10 \pm 0.27$ | $85.73 \pm 0.60$ |
| Mixup | $92.27 \pm 0.19$ | $66.80 \pm 2.84$ | $93.76 \pm 0.19$ | $66.94 \pm 2.00$ | $92.46 \pm 0.44$ | $74.72 \pm 1.09$ |
| EERM | $89.49 \pm 1.55$ | $82.08 \pm 0.61$ | $92.22 \pm 0.37$ | $61.39 \pm 0.85$ | $88.09 \pm 0.55$ | $79.98 \pm 1.28$ |
| DANN | $91.25 \pm 0.37$ | $83.39 \pm 1.87$ | $92.30 \pm 0.77$ | $68.84 \pm 2.53$ | $91.84 \pm 0.79$ | $83.89 \pm 1.37$ |
| CORAL | $91.25 \pm 0.54$ | $84.77 \pm 1.48$ | $92.47 \pm 0.18$ | $62.36 \pm 1.87$ | $91.40 \pm 0.91$ | $85.35 \pm 1.58$ |
| DE | $92.51 \pm 0.00$ | $84.17 \pm 0.00$ | $93.31 \pm 0.00$ | $73.93 \pm 0.00$ | $93.24 \pm 0.00$ | $86.93 \pm 0.00$ |
| ERM + mod | $90.99 \pm 0.16$ | $85.72 \pm 0.34$ | $91.74 \pm 0.24$ | $63.67 \pm 0.35$ | $92.40 \pm 0.34$ | $86.17 \pm 0.20$ |

Table 13: Comparison of several graph methods for improving the OOD robustness in terms of their predictive performance across structural shifts on **AmazonPhoto** dataset.

| | popularity | | locality | | density | |
|---|---|---|---|---|---|---|
| | ID | OOD | ID | OOD | ID | OOD |
| ERM | $95.82 \pm 0.24$ | $87.63 \pm 0.65$ | $93.73 \pm 0.29$ | $64.73 \pm 3.80$ | $94.64 \pm 0.38$ | $91.25 \pm 0.16$ |
| Mixup | $97.99 \pm 0.36$ | $87.73 \pm 2.39$ | $95.55 \pm 0.30$ | $75.36 \pm 2.01$ | $95.50 \pm 0.41$ | $90.71 \pm 2.13$ |
| EERM | $95.90 \pm 0.40$ | $86.84 \pm 1.02$ | $91.95 \pm 0.21$ | $54.06 \pm 0.81$ | $90.46 \pm 0.38$ | $88.91 \pm 0.31$ |
| DANN | $95.25 \pm 0.26$ | $85.23 \pm 0.78$ | $91.81 \pm 0.39$ | $48.79 \pm 5.04$ | $93.99 \pm 0.66$ | $91.87 \pm 0.46$ |
| CORAL | $95.34 \pm 0.27$ | $85.91 \pm 0.67$ | $91.85 \pm 0.51$ | $45.70 \pm 3.34$ | $93.64 \pm 0.80$ | $91.99 \pm 0.43$ |
| DE | $95.82 \pm 0.00$ | $88.82 \pm 0.00$ | $93.59 \pm 0.00$ | $69.15 \pm 0.00$ | $94.38 \pm 0.00$ | $92.09 \pm 0.00$ |
| ERM + mod | $97.07 \pm 0.35$ | $89.95 \pm 0.20$ | $95.01 \pm 0.43$ | $57.41 \pm 1.79$ | $95.22 \pm 0.33$ | $91.97 \pm 0.33$ |

Table 14: Comparison of several graph methods for improving the OOD robustness in terms of their predictive performance across structural shifts on **CoauthorCS** dataset.

| | popularity | | locality | | density | |
|---|---|---|---|---|---|---|
| | ID | OOD | ID | OOD | ID | OOD |
| ERM | $93.60 \pm 0.18$ | $90.67 \pm 0.20$ | $92.34 \pm 0.14$ | $91.22 \pm 0.35$ | $94.30 \pm 0.17$ | $89.71 \pm 0.21$ |
| Mixup | $94.97 \pm 0.33$ | $94.05 \pm 0.38$ | $94.05 \pm 0.13$ | $91.48 \pm 0.16$ | $94.97 \pm 0.29$ | $92.11 \pm 0.15$ |
| EERM | $93.82 \pm 0.08$ | $91.01 \pm 0.15$ | $91.22 \pm 0.27$ | $91.95 \pm 0.21$ | $93.64 \pm 0.23$ | $88.42 \pm 0.15$ |
| DANN | $93.89 \pm 0.26$ | $90.59 \pm 0.12$ | $91.66 \pm 0.37$ | $91.65 \pm 1.29$ | $94.60 \pm 0.16$ | $89.92 \pm 0.29$ |
| CORAL | $93.80 \pm 0.14$ | $90.78 \pm 0.13$ | $91.69 \pm 0.23$ | $91.37 \pm 0.48$ | $94.42 \pm 0.07$ | $89.72 \pm 0.22$ |
| DE | $93.68 \pm 0.00$ | $90.93 \pm 0.00$ | $92.64 \pm 0.00$ | $91.76 \pm 0.00$ | $94.55 \pm 0.00$ | $90.05 \pm 0.00$ |
| ERM + mod | $95.35 \pm 0.15$ | $95.00 \pm 0.06$ | $94.77 \pm 0.09$ | $94.06 \pm 0.11$ | $96.67 \pm 0.11$ | $93.64 \pm 0.16$ |

Table 15: Comparison of several graph methods for improving the OOD robustness in terms of their predictive performance across structural shifts on **CoauthorPhysics** dataset.

| | popularity | | locality | | density | |
|---|---|---|---|---|---|---|
| | ID | OOD | ID | OOD | ID | OOD |
| ERM | $93.60 \pm 0.18$ | $90.67 \pm 0.20$ | $92.34 \pm 0.14$ | $91.22 \pm 0.35$ | $94.30 \pm 0.17$ | $89.71 \pm 0.21$ |
| Mixup | $96.96 \pm 0.11$ | $94.05 \pm 0.74$ | $97.27 \pm 0.11$ | $93.39 \pm 0.95$ | $96.80 \pm 0.08$ | $84.57 \pm 0.85$ |
| EERM | $95.93 \pm 0.10$ | $93.37 \pm 0.08$ | $94.04 \pm 0.16$ | $92.32 \pm 0.11$ | $95.26 \pm 0.03$ | $93.98 \pm 0.06$ |
| DANN | $96.56 \pm 0.14$ | $93.73 \pm 0.25$ | $96.64 \pm 0.23$ | $91.35 \pm 1.12$ | $96.08 \pm 0.17$ | $95.05 \pm 0.11$ |
| CORAL | $96.51 \pm 0.10$ | $93.58 \pm 0.29$ | $96.93 \pm 0.08$ | $86.21 \pm 0.86$ | $95.97 \pm 0.09$ | $95.00 \pm 0.17$ |
| DE | $93.68 \pm 0.00$ | $90.93 \pm 0.00$ | $92.64 \pm 0.00$ | $91.76 \pm 0.00$ | $94.55 \pm 0.00$ | $90.05 \pm 0.00$ |
| ERM + mod | $95.35 \pm 0.15$ | $95.00 \pm 0.06$ | $94.77 \pm 0.09$ | $94.06 \pm 0.11$ | $96.67 \pm 0.11$ | $93.64 \pm 0.16$ |

Table 16: Comparison of several graph methods for uncertainty estimation in terms of their OOD detection performance across structural shifts on **CoraML** dataset.

|  | **Popularity** | **Locality** | **Density** |
|---|---|---|---|
| SE | $75.67 \pm 0.85$ | $87.13 \pm 0.74$ | $81.55 \pm 0.45$ |
| NatPN | $47.31 \pm 9.44$ | $81.66 \pm 3.05$ | $69.05 \pm 3.05$ |
| GPN | $66.95 \pm 1.15$ | $73.09 \pm 2.21$ | $70.95 \pm 1.58$ |
| DE | $70.55 \pm 0.00$ | $93.32 \pm 0.00$ | $84.46 \pm 0.00$ |
| NatPE | $37.59 \pm 0.00$ | $78.48 \pm 0.00$ | $65.83 \pm 0.00$ |
| GPE | $65.67 \pm 0.00$ | $74.23 \pm 0.00$ | $71.07 \pm 0.00$ |
| SE + mod | $55.66 \pm 0.25$ | $72.97 \pm 0.71$ | $68.27 \pm 0.58$ |

Table 17: Comparison of several graph methods for uncertainty estimation in terms of their OOD detection performance across structural shifts on **CiteSeer** dataset.

|  | **Popularity** | **Locality** | **Density** |
|---|---|---|---|
| SE | $68.01 \pm 1.23$ | $89.89 \pm 0.56$ | $66.90 \pm 0.41$ |
| NatPN | $31.18 \pm 1.54$ | $91.96 \pm 3.14$ | $59.77 \pm 1.54$ |
| GPN | $61.99 \pm 1.74$ | $69.53 \pm 5.10$ | $57.41 \pm 1.72$ |
| DE | $56.22 \pm 0.00$ | $98.18 \pm 0.00$ | $70.48 \pm 0.00$ |
| NatPE | $28.07 \pm 0.00$ | $89.02 \pm 0.00$ | $58.10 \pm 0.00$ |
| GPE | $63.01 \pm 0.00$ | $73.89 \pm 0.00$ | $57.94 \pm 0.00$ |
| SE + mod | $54.36 \pm 0.56$ | $78.74 \pm 0.55$ | $61.03 \pm 0.73$ |

Table 18: Comparison of several graph methods for uncertainty estimation in terms of their OOD detection performance across structural shifts on **PubMed** dataset.

|  | **Popularity** | **Locality** | **Density** |
|---|---|---|---|
| SE | $68.60 \pm 0.34$ | $66.34 \pm 1.03$ | $58.60 \pm 0.39$ |
| NatPN | $49.61 \pm 3.17$ | $58.91 \pm 2.89$ | $51.00 \pm 1.45$ |
| GPN | $72.30 \pm 0.29$ | $69.63 \pm 4.84$ | $62.04 \pm 0.68$ |
| DE | $74.31 \pm 0.00$ | $72.19 \pm 0.00$ | $63.04 \pm 0.00$ |
| NatPE | $46.94 \pm 0.00$ | $56.41 \pm 0.00$ | $49.54 \pm 0.00$ |
| GPE | $72.62 \pm 0.00$ | $66.07 \pm 0.00$ | $62.58 \pm 0.00$ |
| SE + mod | $53.68 \pm 0.12$ | $54.86 \pm 0.72$ | $52.25 \pm 0.13$ |

Table 19: Comparison of several graph methods for uncertainty estimation in terms of their OOD detection performance across structural shifts on **AmazonComputer** dataset.

|          | Popularity        | Locality         | Density          |
|----------|-------------------|------------------|------------------|
| SE       | $88.52 \pm 0.37$  | $86.48 \pm 0.80$ | $44.24 \pm 0.57$ |
| NatPN    | $53.83 \pm 15.71$ | $59.92 \pm 6.59$ | $44.42 \pm 3.44$ |
| GPN      | $81.23 \pm 1.44$  | $82.84 \pm 2.56$ | $43.92 \pm 2.56$ |
| DE       | $82.53 \pm 0.00$  | $83.54 \pm 0.00$ | $50.88 \pm 0.00$ |
| NatPE    | $41.33 \pm 0.00$  | $51.93 \pm 0.00$ | $42.09 \pm 0.00$ |
| GPE      | $81.35 \pm 0.00$  | $81.97 \pm 0.00$ | $43.81 \pm 0.00$ |
| SE + mod | $60.54 \pm 0.41$  | $74.09 \pm 0.64$ | $59.27 \pm 0.44$ |

Table 20: Comparison of several graph methods for uncertainty estimation in terms of their OOD detection performance across structural shifts on **AmazonPhoto** dataset.

|          | Popularity        | Locality          | Density           |
|----------|-------------------|-------------------|-------------------|
| SE       | $92.05 \pm 0.27$  | $93.29 \pm 0.60$  | $41.08 \pm 1.89$  |
| NatPN    | $79.72 \pm 7.29$  | $65.66 \pm 11.98$ | $54.88 \pm 17.40$ |
| GPN      | $80.85 \pm 2.64$  | $90.90 \pm 2.82$  | $49.97 \pm 2.24$  |
| DE       | $90.72 \pm 0.00$  | $96.66 \pm 0.00$  | $51.33 \pm 0.00$  |
| NatPE    | $80.43 \pm 0.00$  | $59.74 \pm 0.00$  | $64.69 \pm 0.00$  |
| GPE      | $82.95 \pm 0.00$  | $91.61 \pm 0.00$  | $51.74 \pm 0.00$  |
| SE + mod | $66.93 \pm 0.47$  | $83.09 \pm 1.57$  | $51.36 \pm 0.94$  |

Table 21: Comparison of several graph methods for uncertainty estimation in terms of their OOD detection performance across structural shifts on **CoauthorCS** dataset.

|          | Popularity       | Locality          | Density          |
|----------|------------------|-------------------|------------------|
| SE       | $83.25 \pm 0.40$ | $85.74 \pm 1.06$  | $50.91 \pm 0.26$ |
| NatPN    | $59.99 \pm 5.57$ | $64.53 \pm 12.82$ | $58.47 \pm 8.22$ |
| GPN      | $63.43 \pm 0.46$ | $64.84 \pm 1.57$  | $56.81 \pm 0.54$ |
| DE       | $82.50 \pm 0.00$ | $87.54 \pm 0.00$  | $54.47 \pm 0.00$ |
| NatPE    | $61.00 \pm 0.00$ | $60.55 \pm 0.00$  | $62.63 \pm 0.00$ |
| GPE      | $62.12 \pm 0.00$ | $63.76 \pm 0.00$  | $57.15 \pm 0.00$ |
| SE + mod | $58.05 \pm 0.71$ | $64.43 \pm 1.40$  | $56.99 \pm 0.26$ |

Table 22: Comparison of several graph methods for uncertainty estimation in terms of their OOD detection performance across structural shifts on **CoauthorPhysics** dataset.

|          | Popularity       | Locality          | Density          |
|----------|------------------|-------------------|------------------|
| SE       | $86.60 \pm 0.29$ | $87.73 \pm 0.51$  | $37.50 \pm 0.32$ |
| NatPN    | $56.74 \pm 3.68$ | $59.54 \pm 23.75$ | $58.81 \pm 5.96$ |
| GPN      | $67.48 \pm 0.45$ | $74.05 \pm 2.67$  | $50.99 \pm 0.53$ |
| DE       | $85.94 \pm 0.00$ | $91.93 \pm 0.00$  | $36.15 \pm 0.00$ |
| NatPE    | $56.11 \pm 0.00$ | $46.14 \pm 0.00$  | $56.94 \pm 0.00$ |
| GPE      | $66.29 \pm 0.00$ | $72.16 \pm 0.00$  | $51.89 \pm 0.00$ |
| SE + mod | $64.84 \pm 0.73$ | $88.02 \pm 0.31$  | $44.13 \pm 0.31$ |

# H   Visualization of distributional shifts in graph domain

In Figures 14–20, we provide the visualizations of different structural shifts in graph domain for all the considered graph datasets: ID is blue, OOD is red. Some graphs have multiple connected components — in that case, we keep only the largest one.

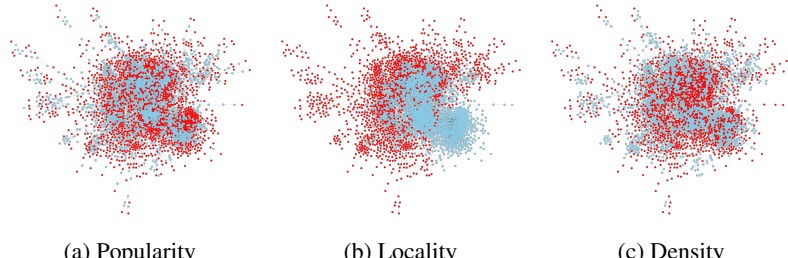

(a) Popularity          (b) Locality          (c) Density

Figure 14: Visualization of structural shifts in graph domain for **CoraML** dataset.

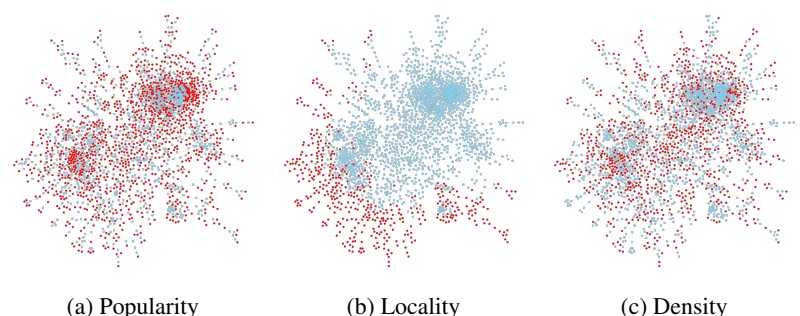

(a) Popularity          (b) Locality          (c) Density

Figure 15: Visualization of structural shifts in graph domain for **CiteSeer** dataset.

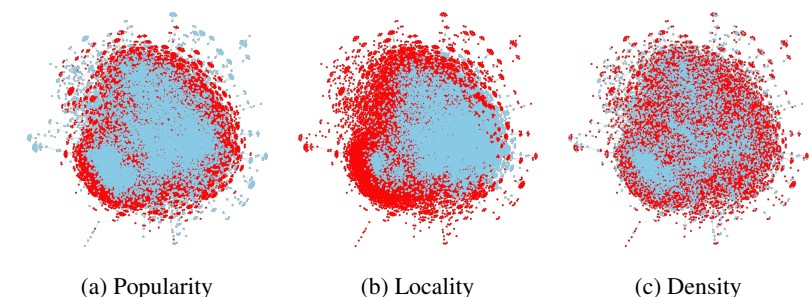

(a) Popularity          (b) Locality          (c) Density

Figure 16: Visualization of structural shifts in graph domain for **PubMed** dataset.

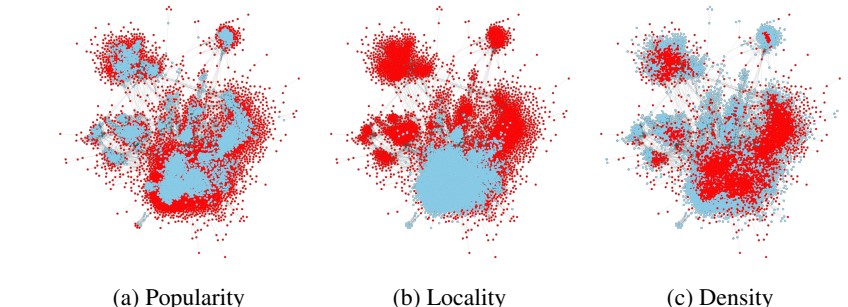

(a) Popularity        (b) Locality        (c) Density

Figure 17: Visualization of structural shifts in graph domain for **AmazonPhoto** dataset.

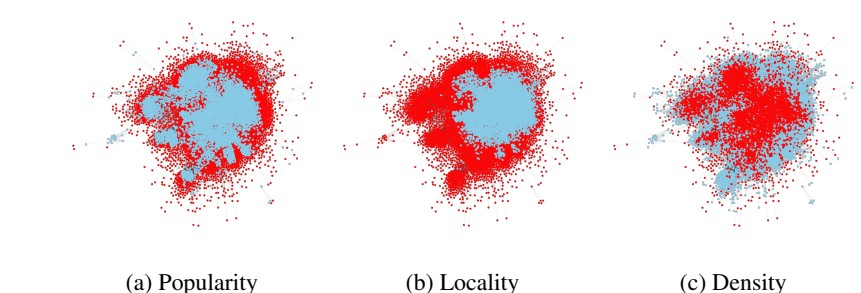

(a) Popularity        (b) Locality        (c) Density

Figure 18: Visualization of structural shifts in graph domain for **AmazonComputer** dataset.

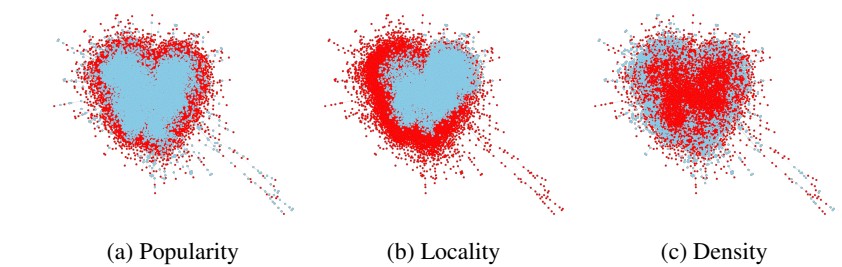

(a) Popularity        (b) Locality        (c) Density

Figure 19: Visualization of structural shifts in graph domain for **CoauthorCS** dataset.

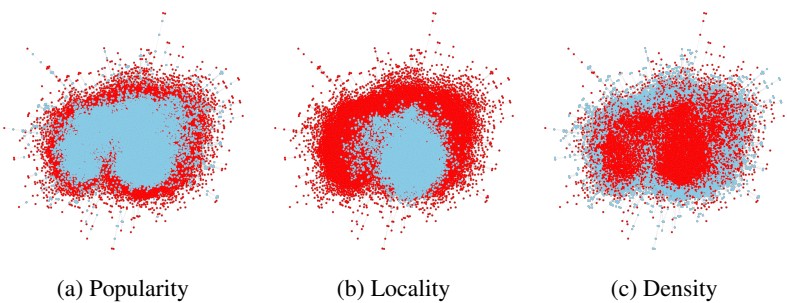

(a) Popularity        (b) Locality        (c) Density

Figure 20: Visualization of structural shifts in graph domain for **CoauthorPhysics** dataset.

# I Visualization of distributional shifts in node feature space

In Figures 21–27, we provide the visualizations of different structural shifts in node feature space for all the considered graph datasets: ID is blue, OOD is red. The pictures are obtained by reducing the original node feature space into the 2D space of t-SNE representations.

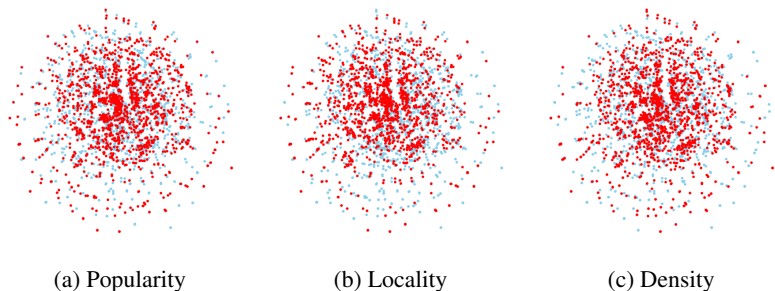

(a) Popularity          (b) Locality          (c) Density

Figure 21: Visualization of structural shifts in node feature space for **CoraML** dataset.

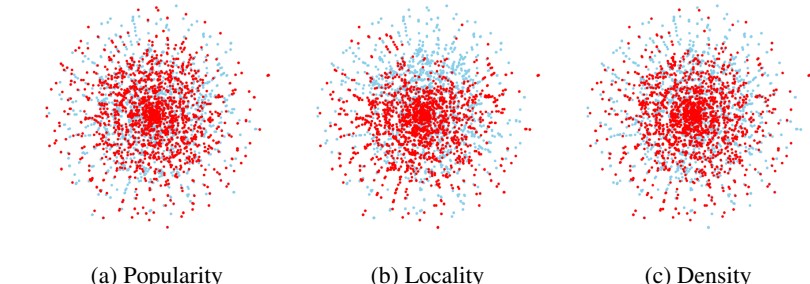

(a) Popularity          (b) Locality          (c) Density

Figure 22: Visualization of structural shifts in node feature space for **CiteSeer** dataset.

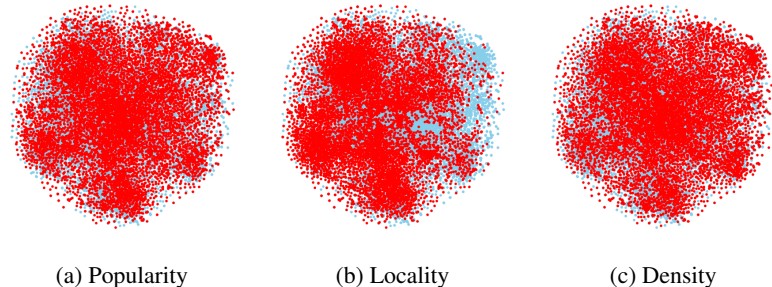

(a) Popularity          (b) Locality          (c) Density

Figure 23: Visualization of structural shifts in node feature space for **PubMed** dataset.

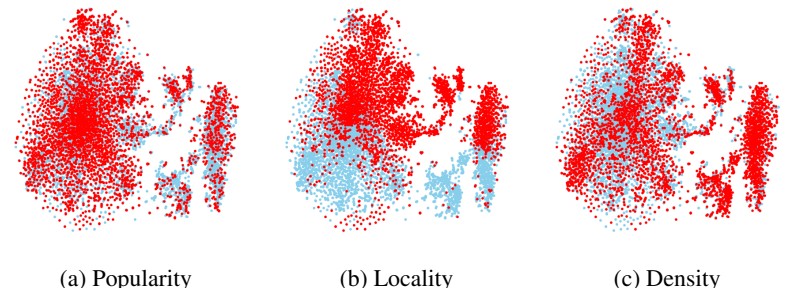

(a) Popularity        (b) Locality        (c) Density

Figure 24: Visualization of structural shifts in node feature space for **AmazonPhoto** dataset.

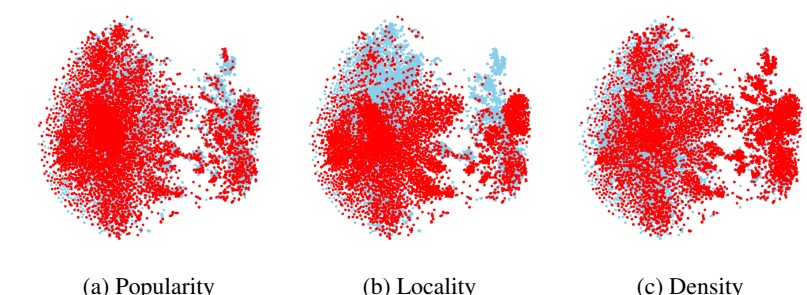

(a) Popularity        (b) Locality        (c) Density

Figure 25: Visualization of structural shifts in node feature space for **AmazonComputer** dataset.

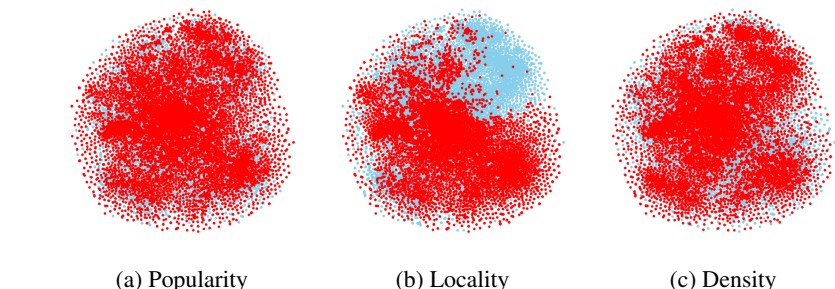

(a) Popularity        (b) Locality        (c) Density

Figure 26: Visualization of structural shifts in node feature space for **CoauthorCS** dataset.

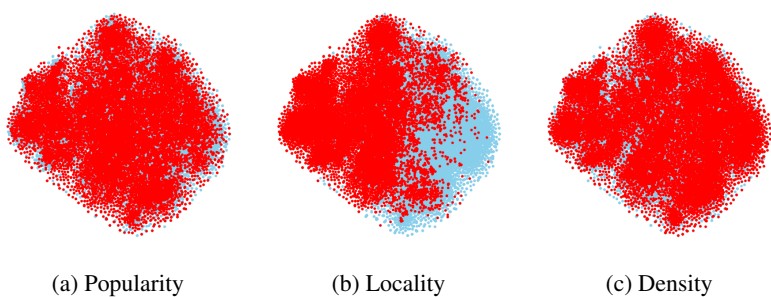

(a) Popularity        (b) Locality        (c) Density

Figure 27: Visualization of structural shifts in node feature space for **CoauthorPhysics** dataset.

