# OpenReview forum: "Evaluating Robustness and Uncertainty of Graph Models Under Structural Distributional Shifts"
_NeurIPS.cc/2023/Conference — NeurIPS 2023 poster_

### Official Review · Reviewer_2ynW · 2023-06-29

**Soundness:** 2 fair
**Presentation:** 3 good
**Contribution:** 2 fair
**Rating:** 4
**Confidence:** 4

**Summary:**

This paper proposes a benchmark for evaluating model robustness against structural distributional shifts. The authors propose three different metrics to split the datasets for the evaluation. In the experiments, the authors evaluate different models under the proposed distributional shifts and show that they can be quite challenging for existing graph models.

**Strengths:**

1. This paper targets an understudied problem of evaluating model robustness and uncertainty estimation under structural distribution shift.

2. It presents a novel idea of using different evaluation metrics to create data splits for structural distribution shifts.

3. The authors evaluate different methods based on the proposed metrics and show some empirical findings.

**Weaknesses:**

1. The authors consider less important nodes are OOD nodes. However, real-world graphs follow the power-law distribution and only have a small number of nodes that have a high degree/node importance. So the proposed metric is kind of counterintuitive to me.

2. The experimental setup, particularly the equal splitting of in-distribution (ID) and out-of-distribution (OOD) data, raises concerns about the generalizability of the findings. Real-world scenarios typically involve a smaller proportion of OOD data, and investigating varying ratios of ID to OOD data would enhance the practical relevance of the evaluation.

3. Since the authors argue the existing methods only focus on feature distribution shift, while this paper only focuses on the structural distribution shift. It would be better to consider them together rather than separately evaluate them, as they often occur simultaneously and have interdependencies.

4. While the proposed metrics for creating data splits are new, the authors could provide more detailed justification and analysis for their choices. It would be beneficial to explore the sensitivity of the evaluation results to different metric configurations.

5. I have a concern that should this paper be submitted to the datasets and benchmark datasets since this paper doesn't focus on technical contribution.

To strengthen the paper, it would be beneficial to address these additional points and provide more comprehensive justifications, analyses, and experiments.

**Questions:**

In addition to the above weaknesses, another question is (line 290-292), the authors"... including DANN, CORAL, EERM, and Mixup, we use the experimental framework from the GOOD benchmark, whereas the remaining methods are implemented in our custom experimental
framework ...", what does it mean? Why not just use the same experimental framework?

---

> ### Author Rebuttal · Authors · 2023-08-09
>
> Thanks for your review! We appreciate that you also consider the problem of evaluating robustness and uncertainty of graph models under structural distributional shift to be understudied, and that you find our idea to use structural node properties for inducing distributional shifts novel. Below, we address your concerns and questions.
>
> W1. If we understand correctly, you consider here another notion of OOD that is based on how many objects in the given dataset have a particular structural property and suggest that the observations with the rarest values of a structural property should be treated as OOD (for instance, as you mentioned, the nodes with high degree in the power-law distribution should belong to OOD). At the same time, rather than focusing on which proportions of nodes share some specific structural property, we try to construct OOD based on how labels might be assigned to the nodes in real-world applications (and in what order they become available for a graph model to extract knowledge from them). Even if there are much more unimportant nodes in the network, they can easily appear to be OOD for a graph model that has learned the structural patterns of the most important nodes that had been investigated first. So, your remark about the power-law distribution of node degrees in real-world networks is true, but it does not contradict our approach to model structural shifts.
>
> W2. Thank you for pointing this out. We believe that in most real-world scenarios, the presence of a large amount of labeled data is rather an exception. In our empirical study, we try to simulate such a situation when the lack of a sufficient amount of labeled data is combined with the presence of structural distributional shift.
>
> Nevertheless, we agree with the importance of investigating other ratios of ID and OOD data to understand whether our findings can be transferred to other cases. Because of that, we have conducted an additional series of experiments where we change the proportion of ID and OOD data, and we discuss them in our [general response](https://openreview.net/forum?id=nJFJcgjnGo&noteId=EvXe6dwLzG).
>
> W3. As you have mentioned, the problem of evaluating model robustness and uncertainty estimation under structural distribution shift is under-explored. In this work, we focus on this aspect and isolate ourselves from the task of simulating complex distributional shifts that are explicitly based on both the structural properties and the node features. It facilitates our study of existing graph models and investigation of various effects in their performance associated with the changes of graph structure. Please also note that graph structure is the only common modality of different graph datasets that can be explored in the same manner, whereas node features may vary significantly across different domains and applications.
>
> However, we have conducted additional analysis to show that our proposed structural shifts also induce the distributional shift in feature space of the considered graph datasets. Several figures demonstrating this effect can be found in our [general response](https://openreview.net/forum?id=nJFJcgjnGo&noteId=EvXe6dwLzG). As future research, it would be interesting to consider some techniques for inducing distributional shifts in graph data where the changes in graph structure and node features could be controlled simultaneously.
>
> W4. Our main idea is that data splits can be created based on various structural properties. Thus, one can potentially use any other node characteristics within our split strategy. In our experiments, we have chosen three particular options that are diverse and reasonable: please, refer to Section 3.2 for a detailed motivation of our choices. In Section 5.1, we investigate how challenging the proposed structural shifts are for some existing graph models in terms of their robustness and uncertainty estimation. Moreover, Appendix A demonstrates how these shifts affect other important graph properties, such as class balance, degree distribution, and pairwise node distances. Finally, in Appendices B and C, we discuss how our approach to create distributional shifts, along with three particular choices of structural node properties, allow us to create a more challenging setting for evaluating the performance of graph models, while complementing and extending the existing techniques. We hope that these sections make the analysis of the proposed structural properties fairly detailed and complete. However, if you have any ideas on how we can further improve this part of our study, could you please share your thoughts so we can adjust the paper accordingly?
>
> W5. Please refer to our [general response](https://openreview.net/forum?id=nJFJcgjnGo&noteId=EvXe6dwLzG).
>
> Q1. In this sentence, we clarify that the experiments with the methods for improving OOD robustness, where we measure the predictive performance in standard node classification, are conducted using the GOOD framework, as it provides all necessary functions. However, the experiments with uncertainty estimation methods, which are evaluated in OOD detection, required a substantially different set of functions (e.g., basic routines for predictions, architecture design, evaluation procedure, metrics, etc.), so we found it much easier to develop a custom experimental framework rather than extending that of GOOD benchmark.

---

> > ### Comment · Reviewer_2ynW · 2023-08-16
> >
> > Thanks for the response, part of my concerns have been addressed. I would like to consider raising my score if the following questions can be answered.
> >
> > Regarding W2, I appreciate that the authors provide a new analysis on the setting with 70% ID, and 30% OOD, but 30% OOD is still a lot, if it is possible to show the prediction/detection performance for 5% or 10% OOD data? And what would be findings?
> >
> > For W3, the authors show that the structure distribution shifts can induce the feature distribution shift as shown in Appendix (Figure 1). It's a bit unclear to me by saying "reducing the original node feature space into the 2D space of t-SNE representations". Why the structural shift will affect the original node features? Or you use models such as GNN to learn the embeddings and then do T-SNE.
> >
> > My last concern is about the contribution/insights of this paper. This paper proposes multiple ways to create structure shifts and evaluate the performance of existing methods on it. We know it is quite challenging for existing methods, but I would expect the authors to propose a solution (could be straightforward) for solving the problem as well, or at least provide the insights or design principles for addressing the structural distribution shifts.

---

> > > ### Author Response · Authors · 2023-08-18
> > >
> > > Thank you for your feedback and involvement in the discussion!
> > >
> > > > … if it is possible to show the prediction/detection performance for 5% or 10% OOD data? And what would be findings?
> > >
> > > Following your suggestion, we have conducted an additional series of experiments where the ID to OOD ratio is 90% to 10%. Our findings are the following:
> > >
> > > The drop in predictive performance under the popularity-based and locality-based shifts reaches almost 7% instead of 3% in the previous setup (where we had 70% to 30% ratio) and 10% instead of 6%, respectively. This was expected and can be explained by the fact that graph models are now tested on the nodes with the most extreme structural properties (i.e., nodes with the lowest PageRank or clustering coefficient), and their inaccurate predictions are no longer compensated by the more accurate ones, which were made on the nodes with far less anomalous properties.
> > >
> > > At the same time, the performance drop on locality-based shift appears to be no more than 1% on average. This result is quite reasonable, since now graph models have access to the whole variety of graph substructures and node features, which allows them to predict equally well for any graph region regardless of its distance to some particular node. The observed effects are interesting and also important to consider when using our approach for evaluating graph models.
> > >
> > > Regarding the OOD detection performance, we observe that results on the locality-based shift keep decreasing and reach 72 points, in contrast to 78 points in the previous setup (and this might happen for the same reason as we discussed the rebuttal). At the same time, the detection metrics on the remaining structural shift appear to be nearly the same (80 points instead of the previous 81 on the popularity-based shift) or even better on average (62 points instead of 56 on the density-based shift), which is consistent with the changes in predictive performance discussed above.
> > >
> > > As for the ranking of baseline models, it remains almost the same, with the most simple methods providing top performance on the majority of prediction tasks.
> > >
> > >
> > > > … It's a bit unclear to me by saying "reducing the original node feature space into the 2D space of t-SNE representations"
> > >
> > > We apologize for causing misunderstanding in our response. In Figure 1, we have just applied t-SNE to the original node features (i.e., we did not use any supervised GNN embeddings) to visualize the distribution of multi-dimensional node features in 2D. This Figure shows that the considered structural shift not only affects the graph structure, but may also lead to a distributional shift in the node features, although the significance of this effect may vary between graph datasets. Our finding is natural, as node features and graph structure are correlated in real-world graphs.
> > >
> > > > … but I would expect the authors to propose a solution (could be straightforward) for solving the problem as well, or at least provide the insights or design principles for addressing the structural distribution shifts.
> > >
> > > Regarding your last question, we partially cover this in Section 5.3 and demonstrate that it is possible to make several simple modifications to the GNN model which help it to deal with distributional shifts.
> > >
> > > Additionally, we have conducted several experiments with data augmentation techniques that were expected to improve the performance of graph models. In particular, we tried using DropEdge or stacking the original node features with the structural node characteristics that correspond to particular distributional shifts. However, we have not achieved consistent improvements in these experiments.
> > >
> > > In summary, this is a very important question, but a principled solution to it is non-trivial and requires further investigation. We believe that our proposed method for creating distributional shifts will support studies in this area.
> > >
> > > We hope that our response answers your questions and we are open to further discussion on the raised points.

---

> > > > ### Comment · Reviewer_2ynW · 2023-08-21
> > > >
> > > > Thanks for your response, I have raised my score to 4.

---

### Official Review · Reviewer_E4Gb · 2023-07-03

**Soundness:** 3 good
**Presentation:** 4 excellent
**Contribution:** 3 good
**Rating:** 7
**Confidence:** 5

**Summary:**

The authors propose three domain selections, namely, popularity, locality, and density to equip node prediction datasets with structural distribution shifts. Extensive experiments are conducted to compare present OOD generalization methods and OOD detection methods.

**Strengths:**

S1: The proposed distribution shifts are novel.

S2: The paper is well written and easy-to-follow.

S3: The evaluations are clear and sufficient.

**Weaknesses:**

W1: The title is misleading. Since this benchmark only focuses on node prediction tasks, the title should reflect it. As we know, structural distribution shifts are different in graph-level and node-level tasks.

W2: The chosen shifts are somehow limited because they are not extracted from real-world scenarios.

W3: Line 39: "Also, the splitting strategies in GOOD are mainly based on the node features and do not take into account the graph structure" is somehow misleading. I suggest the authors to clarify that "GOOD mainly focuses on node feature shifts in **node-level tasks**".


**Questions:**

Q1: Can you find a way to compare your proposed shifts with the real shifts?

Q2: I would suggest submitting benchmark works to the Datasets and Benchmarks track, which offers more transparent evaluations to your benchmark.

**Limitations:**

Broader impacts are expected to be discussed.
The licenses of the datasets and the code (GOOD) are expected to be mentioned.

---

> ### Author Rebuttal · Authors · 2023-08-09
>
> Thanks for your review! We appreciate your encouraging feedback about our work and address your comments below.
>
> W1. Thanks for this remark. Indeed, the notion of structural distributional shift might be significantly different in graph-level problems, so it is important to emphasize that our approach to induce structural shifts is primarily intended for node-level problems. We will change the title accordingly.
>
> W2. This concern is reasonable. Because of that, we devote a significant part of the text to the discussion of this limitation. In Section 3.2, we provide the motivation behind the proposed structural shifts — specifically, we describe how they can arise in real-world applications and how they can be modeled by our split strategies based on various structural node properties. Of course, having real distribution shifts for a particular application would be preferable, but our goal was to address the situations when such natural splits are unavailable. Moreover, in Appendix C, we show that the proposed structural shifts enable us to catch some properties of the distributional shifts between train and test subsets in realistic data splits of the OGB benchmark. Note that we outline our limitations in a separate paragraph of conclusion.
>
> W3. Thanks for this comment, we will make the necessary clarification in the revised version of our paper.
>
> Q1. Yes, this is done to some extent in Appendix C, where we discuss how the chosen node-level characteristics that describe particular structural properties, including PageRank, PPR and clustering coefficient, correlate with the distributional shifts induced by the split strategies of the OGB benchmark. Unfortunately, there are not many examples of real distributional shifts in known graph benchmarks. Otherwise, we would not need such an approach that allows us to create data splits with synthetic structural shifts that replicate the properties of the real ones.
>
> Q2. Please refer to our [general response](https://openreview.net/forum?id=nJFJcgjnGo&noteId=EvXe6dwLzG).

---

> > ### Comment · Reviewer_E4Gb · 2023-08-11
> >
> > Thank you for your replies.
> >
> > Most of my concerns are addressed. And I also noticed that some of them cannot be addressed in this single work, so please list them and discuss them in the limitations and further works extensively. Although this is not a paper on the dataset and benchmark track, I have to clarify that my current score of 7 is partially based on the trust that the authors can be responsible and responsive for their released code, which is *essential even though the authors claim that it is not a traditional benchmark paper*.
> >
> > Furthermore, I'm willing to defend my review: Although I acknowledge that the suggestions of Reviewer 2ynW are beneficial, many concerns proposed are **not major** to reject this paper, since I don't think these concerns can undermine the contributions and conclusions of this paper.
> >
> > Score: $6\rightarrow 7$: Accept.

---

> > > ### Author Response · Authors · 2023-08-18
> > >
> > > Thanks for your support! We plan to extend the discussion of future work based on your questions. Our code will be supported in our open source repository and also in the most popular frameworks for deep learning on graphs: PyG and DGL.

---

### Official Review · Reviewer_gSyv · 2023-07-05

**Soundness:** 3 good
**Presentation:** 3 good
**Contribution:** 2 fair
**Rating:** 6
**Confidence:** 4

**Summary:**

This paper provides a benchmark solution for producing diverse and meaningful distributional shifts from existing graph datasets. The experiments conducted demonstrate that the proposed distributional shifts can be difficult for existing graph models, and surprisingly, simpler models often outperform more sophisticated ones. The experiments also reveal a trade-off between the quality of learned representations for the base classification task and the ability to distinguish nodes from different distributions using these representations.

**Strengths:**

The proposed data-splitting approach is quite interesting. The overall presentation is quite easy to follow.
Based on the provided data-splitting strategy, most graph/node classification methods may not achieve a good classification performance and uncertainty quantification quality. It is also nice to see the authors conduct many empirical studies to validate the idea and compare the difference between various baselines.

**Weaknesses:**

1. The technical novelty is somewhat limited. It seems like the technical contribution is based on existing techniques (PageRank, PPR, ...). I would like to see more discussions of the rationale/motivation for choosing the specific metric. For example, the PageRank is selected because of the need to quantify the node popularity, but other node centrality-based metrics can also be used to quantify the node popularity.
2. The dataset size is also quite small. It's not promising the experience learned from these small networks can be extended to million-scale networks.
3. Some important baselines (e.g., GOOD: A Graph Out-of-Distribution Benchmark) are not included in the main context.

I feel like this paper should be more suitable for the dataset and benchmark track.

**Questions:**

Please refer to the Weakness section.

**Limitations:**

 I am not seeing limitations of this work.

---

> ### Author Rebuttal · Authors · 2023-08-09
>
> Thanks for your review! We appreciate that you find our approach to create data splits interesting and the presentation easy to follow. Below, we reply to your comments.
>
> W1. Our main contribution is an approach for creating data splits with distributional shifts based on various structural properties. For our experiments, we consider some particular examples of node characteristics (specifically, PageRank, PPR, and clustering coefficient) as they are quite diverse, widely adopted in the community, and related to situations that may arise in practice. However, as mentioned in Section 3.2, other node characteristics can also be used instead of the chosen ones (e.g., node degree as the measure of popularity or graph distance as the measure of locality). As for PageRank, we choose this metric because it is less discrete than node degree and also widely used in the literature.
>
> W2. We conduct our experiments on several graph datasets that have been used in most previous works and adopted in established graph benchmarks. They come from various domains and applications, while having different structure and number of node features. Although these datasets do not fall under the category of large-scale datasets, they still enable us to conduct the necessary experiments and support our claims about the performance of graph models under structural distributional shifts.
>
> However, if you feel that our discussion can remain incomplete without some large-scale graph datasets, please let us know which of them we should include, and we will do our best to conduct additional experiments on them and include our results into the revised version.
>
> W3. Could you please explain what you mean by such baselines?
> * If you would like to see the evaluation of other methods for improving OOD that have been considered in the GOOD benchmark and have some particular recommendations, please let us know the names of these methods so that we could test them on the proposed structural shifts and include the results into the revised version;
> * If your comment refers to the lack of comparison between our distributional shifts and those proposed in the GOOD benchmark, we can extend the main text with the content of Appendix B, which describes how our approach to create structural shifts extends GOOD and allows us to overcome some of their practical limitations.
>
> > I feel like this paper should be more suitable for the dataset and benchmark track.
>
> Please refer to our [general response](https://openreview.net/forum?id=nJFJcgjnGo&noteId=EvXe6dwLzG).

---

> > ### Comment · Reviewer_gSyv · 2023-08-17
> > **Rebuttal Acknowledgement**
> >
> > Thanks for the rebuttal. Part of my concerns have been addressed, I would still like to raise my score from 5 to 6.
> >
> > However, my concerns about the network size and potential impact still remain. In terms of network size, the large scale typically means million scales of node numbers, but the largest network used in this work is CoauthorPhysics (34493, 495924), which is far from large-scale.

---

> > > ### Author Response · Authors · 2023-08-18
> > >
> > > Thanks for your positive feedback! Following your suggestion, we plan to conduct experiments on ogbn-products, an OGB dataset that contains more than 2M nodes, and include them in the revised version.

---

### Official Review · Reviewer_AZW5 · 2023-07-06

**Soundness:** 3 good
**Presentation:** 3 good
**Contribution:** 3 good
**Rating:** 7
**Confidence:** 5

**Summary:**

This paper proposes a general approach for inducing diverse distributional shifts based on graph structure and evaluates the robustness and uncertainty of graph models under these shifts. The authors define several types of distributional shifts based on graph characteristics, such as popularity and locality, and show that these shifts can be quite challenging for existing graph models. They also find that simple models often outperform more sophisticated methods on these challenging shifts. Additionally, the authors explore the trade-offs between the quality of learned representations for the base classification task and the ability to separate nodes under structural distributional shift. Overall, the paper's contributions include a novel approach for creating diverse and challenging distributional shifts for graph datasets, a thorough evaluation of the proposed shifts, and insights into the trade-offs between representation quality and shift detection.

**Strengths:**

Originality: The paper's approach for inducing diverse distributional shifts based on graph structure is novel and fills a gap in the existing literature, which has mainly focused on node features. The authors' proposed shifts are also motivated by real-world scenarios and are synthetically generated, making them a valuable resource for evaluating the robustness and uncertainty of graph models.

Quality: The paper's methodology is rigorous and well-designed, with clear explanations of the proposed shifts and the evaluation metrics used. The authors also provide extensive experimental results that demonstrate the effectiveness of their approach and highlight the challenges that arise when evaluating graph models under distributional shifts.

Clarity: The paper is well-written and easy to follow, with clear explanations of the proposed shifts and the evaluation methodology. The authors also provide helpful visualizations and examples to illustrate their points.

Significance: The paper's contributions are significant and have implications for the development of more robust and reliable decision-making systems based on machine learning. The authors' approach for inducing diverse distributional shifts based on graph structure can be applied to any dataset, making it a valuable resource for researchers and practitioners working on graph learning problems. The insights into the trade-offs between representation quality and shift detection are also important for understanding the limitations of existing graph models and developing more effective ones. Overall, the paper's contributions have the potential to advance the field of graph learning and improve the reliability of machine learning systems.

**Weaknesses:**

Limited scope: The paper focuses solely on node-level problems of graph learning and does not consider other types of graph problems, such as link prediction or graph classification. This limited scope may restrict the generalizability of the proposed approach and its applicability to other types of graph problems.

Synthetic shifts: While the authors' proposed shifts are motivated by real-world scenarios, they are synthetically generated, which may limit their ability to capture the full complexity of real distributional shifts. The paper acknowledges this limitation, but it is still worth noting that the proposed shifts may not fully reflect the challenges that arise in real-world scenarios.

Limited comparison to existing methods: While the paper provides extensive experimental results that demonstrate the effectiveness of the proposed approach, it does not compare the proposed approach to other existing methods for evaluating graph models under distributional shifts. This limits the ability to assess the relative strengths and weaknesses of the proposed approach compared to other approaches.

**Questions:**

Here are some questions and suggestions for the authors:

Can the proposed approach be extended to other types of graph problems, such as link prediction or graph classification? If so, how might the approach need to be modified to accommodate these different types of problems?

How might the proposed approach be adapted to handle real-world distributional shifts, rather than synthetically generated shifts? Are there any limitations to the proposed approach that might make it less effective in handling real-world shifts?

How does the proposed approach compare to other existing methods for evaluating graph models under distributional shifts? Are there any specific strengths or weaknesses of the proposed approach compared to other approaches?

The paper notes that there is a trade-off between the quality of learned representations for the target classification task and the ability to detect distributional shifts using these representations. Can the authors provide more details on this trade-off and how it might impact the effectiveness of the proposed approach in different scenarios?

The paper proposes several different types of distributional shifts based on graph structure. Can the authors provide more details on how these shifts were chosen and whether there are other types of shifts that might be relevant for evaluating graph models?

The paper focuses on evaluating graph models under distributional shifts, but does not provide any guidance on how to modify existing models to improve their robustness to these shifts. Can the authors provide any suggestions or guidelines for modifying existing models to improve their performance under distributional shifts?

The paper notes that the proposed approach can be applied to any dataset, but does not provide any guidance on how to choose the appropriate node property to use as a splitting factor. Can the authors provide any suggestions or guidelines for choosing an appropriate node property for a given dataset?

Overall, the paper presents an interesting and novel approach for evaluating graph models under distributional shifts. However, there are several areas where the authors could provide more details or guidance to help readers better understand and apply the proposed approach.

**Limitations:**

The paper briefly acknowledges some of the limitations of the proposed approach, such as the fact that the synthetic shifts may not fully reflect the complexity of real-world distributional shifts. However, the paper does not provide a detailed discussion of the potential negative societal impact of the work.

Given the technical nature of the paper, it is possible that the authors did not see a direct connection between their work and potential negative societal impacts. However, it is always important for authors to consider the broader implications of their work, especially in fields like machine learning where there is a growing awareness of the potential risks and harms associated with these technologies.

In future work, the authors could consider providing a more detailed discussion of the potential societal impacts of their work, including any ethical or social considerations that may arise from the use of their approach. This could help to ensure that the work is being developed and applied in a responsible and ethical manner.

---

> ### Author Rebuttal · Authors · 2023-08-09
>
> Thanks for your review! We appreciate your support and address your comments below (the order of questions is preserved).
>
> Q1. In graph-level problems, the notion of structural distributional shift might be significantly different and depend on particular applications. On the contrary, many node-level and edge-level problems have a lot in common in terms of what a plausible structural shift may look like. For instance, in link prediction problems, one may have to train their model on the edges that connect very popular nodes to unpopular ones, which is a very common and meaningful situation in real-world scenarios, and then test on the edges between mostly unpopular nodes. The same sort of reasoning can be done for the structural shifts based on locality, where one has access only to some local region of a graph and need to predict edges between the nodes of a distant region. Thus, our proposed structural shifts can be easily transferred for evaluating the robustness and uncertainty of graph models in edge-level problems.
>
> Q2. Our approach is designed specifically for situations when real-world distributional shifts are not available. In such cases, synthetically generated shifts can serve as a reasonable approximation. In Appendix C, we show that the proposed structural shifts enable us to catch some properties of real-world ones.
>
> Q3. Please refer to Appendix B, where we discuss how our method for creating data splits with structural shifts extends the GOOD benchmark, and how it allows us to overcome some of their practical limitations. Moreover, in Appendix C, we show that the proposed structural shifts enable us to catch some properties of the distributional shifts between train and test subsets in realistic data splits of the OGB benchmark. If these sections do not cover some important aspects of comparison, please let us know.
>
> Q4. We note that our approach for generating distributional shifts is orthogonal to the problem of developing models capable of dealing with them. We hope that our diverse and complex structural shifts will help researchers in investigating this trade-off and developing models that are both robust to distributional shifts and able to detect them.
>
> Q5. This question is covered in Section 3.2, where we discuss the motivation behind our proposed structural shifts and how they can arise in real-world applications. In addition, we mention there some other examples of node-level characteristics that can be used to describe the chosen structural properties, such as node degree as the measure of popularity and graph distance as the measure of locality. The structural shifts considered in our paper are intuitive and practically motivated. However, any node characteristic can be used for splitting, and also different characteristics can potentially be combined to create more complicated shifts.
>
> Q6. In Section 5.3, we partially cover this question and demonstrate that it is possible to make several modifications to the GNN model which help it to deal with distributional shifts. However, a more principled solution to this question is non-trivial and requires further investigation. We hope that our proposed method for creating distributional shifts will support studies in this area.
>
> Q7. In fact, the choice of some structural node property as a splitting factor might depend significantly on a particular application. In our paper, we consider quite diverse node characteristics that have different motivations behind them and impact on the performance of graph models, so we recommend using all of them to conduct experiments and aggregate the obtained results for making final decisions.

---

### Author Rebuttal · Authors · 2023-08-09

**General response**

We would like to thank all the reviewers for their valuable feedback and suggestions! In this general response, we address questions raised by several reviewers and describe additional experiments we conducted.

**Additional experiments**

As requested by Reviewer 2ynW, we extend our empirical study with greater ID to OOD ratio (70% to 30% instead of 50% to 50% as in the paper) and provide the results in the attached PDF.

Let us revisit our analysis of the proposed structural shifts. As can be seen in Table 1, the results in both OOD robustness, which is evaluated in node classification task, and OOD detection, which is done by means of uncertainty estimation, have not changed much compared to the previous setup — the considered graph models show the average drop in performance of 3% on the popularity-based shift and 6% on the density-based shift, while the OOD detection performance remains at nearly 81 and 56 points, respectively. Despite the fact that the graph models now have access to more diverse structures in terms of popularity and local density, making decisions about unimportant and sparsely surrounded nodes remains to be as difficult as before.

Interestingly, the drop in predictive performance on the locality-based shift appears to be less significant compared to the original setup and reaches only 5% on average instead of the previous 15%. At the same time, the OOD detection performance on this structural shift drops to 78 points, whereas it was 85 points in the original setup. These results also match our intuition — as the distance between the ID and OOD nodes decreases, the OOD samples become less distinguishable from the ID ones. This makes the OOD detection performance drop on average, while the gap between ID and OOD metrics in standard node classification disappears.

Now we may compare the existing graph models based on the results in Table 2. As can be seen, the ranking of methods remains almost the same as in the original setup. Specifically, the most simple data augmentation technique Mixup often shows the best performance in OOD robustness, while DE provides the second best results on most structural shifts. As for OOD detection, the methods based on the entropy of predictive distribution again outperform the Dirichlet ones, and the uncertainty estimates produced by DE are best correlated with the OOD examples. Thus, our observations regarding the performance of graph models are consistent with those reported in the paper.

**Additional analysis**

As a response to Reviewer 2ynW, we have also conducted a brief analysis of distributional shift in feature space induced by the proposed structural shift. In particular, we used t-SNE to embed the original feature space of the considered graph datasets into 2D representation space. In Figure 1 of the attached PDF, one can see the examples of such a visualization for the locality-based structural shift that also creates a notable distributional shift in node features. It supports our reasoning in Appendix B that a realistic shift most commonly impacts both node features and graph structure. In the revised version of our paper, we will include the figures for other graph datasets and shifts.

**On submission track**

Several reviewers expressed their doubts about whether our work should be at the main conference track or at the Datasets and Benchmarks (DB) track. Let us explain our choice.

According to the DB track description, its main purpose is to present new datasets to the community, with a special focus on their maintenance and accessibility via a properly designed API. Among other things, a submission to the DB track requires the information about how the data is collected and organized, what kind of information it contains, how it should be used ethically and responsibly, etc.

On the contrary, our work is about a new approach to induce structural distributional shifts that can be applied to any graph dataset. In other words, our work presents a way of looking at and dealing with data, rather than a new source of data. Further, according to the call for papers, our work fits the scope of the main track. It is written in the conference FAQ that if a paper fits both tracks, it remains the authors' choice where to submit it. Taking into account the above arguments, we decided to submit to the main track.

**Broader impact**

As requested by the reviewers, we will add a discussion of broader impact to our paper. In particular, we assume that the proposed approach for evaluating robustness and uncertainty of graph models will support the development of more reliable systems based on machine learning. By testing on the presented structural shifts, it should be easier to detect various biases against under-represented groups that may have a negative impact on the resulting performance and interfere with fair decision-making.

---

### Decision · Program_Chairs · 2023-09-21

**Decision:**

Accept (poster)

**Comment:**

This paper proposes a method for inducing structural distributional shifts that can be applied to graph datasets and help to explore robustness of various algorithms with respect to such shifts. The reviewers found the paper to be novel and interesting, with potential implications for research  studying the robustness of node prediction algorithms. While there were also some concerns such as the size of the graphs used in the experiments, the overall consensus was that the paper has sufficient value and should be accepted to the conference.